

# LIVVkit 2.1: Automated and extensible ice sheet model validation

Katherine J. Evans[1], Joseph H. Kennedy[1], Dan Lu[1], Mary M. Forrester[2], Stephen Price[3], Jeremy Fyke[3,a],
Andrew R. Bennett[4], Matthew J. Hoffman[3], Irina Tezaur[5], Charles S. Zender[6], and Miren Vizcaíno[7]

[1]Computational Earth Sciences Group, Oak Ridge National Laboratory, Oak Ridge, TN, USA
[2]Colorado School of Mines, Golden, CO, USA
[3]Fluid Dynamics and Solid Mechanics Group, Los Alamos National Laboratory, Los Alamos, NM, USA
[4]U. Washington Dept. of Civil and Environmental Engineering
[5]Sandia National Laboratories, Albuquerque, NM, USA
[6]Departments of Earth System Science and Computer Science, University of California, Irvine, USA
[7]Institute for Marine and Atmospheric Research, Utrecht University, Utrecht, Netherlands
[a]now with Associated Engineering, Vernon, BC, Canada

**Correspondence:** K. J. Evans (evanskj@ornl.gov)

**Abstract.** A collection of scientific analyses, metrics, and visualizations for robust validation of ice sheet models is presented
using the LIVVkit package, version 2.1. This software collection targets stand-alone ice sheet or coupled Earth system models,
and handles datasets and operations that require high-performance computing and storage. LIVVkit aims to enable efficient
and fully reproducible workflows for post-processing, analysis, and visualization of observational and model-derived datasets
5  in a shareable format, whereby all data, methodologies, and output are distributed to users for evaluation. We demonstrate
LIVVkit validation for a Greenland ice sheet simulation using the coupled Community Earth System Model, CESM, as well as
an idealized stand-alone high-resolution ice sheet model, CISM-Albany. As one example of the capability, LIVVkit analyzes
the degree to which models capture the surface mass balance (SMB) and identifies potential sources of bias, using recently
available in-situ and remotely sensed data as comparison. Related fields within atmosphere and land surface models, e.g.
10  surface temperature, radiation, and cloud cover, are also diagnosed. Applied to the CESM1.0, LIVVkit identifies a positive
SMB bias that is focused largely around Greenland's southwest region that is due to insufficient ablation.

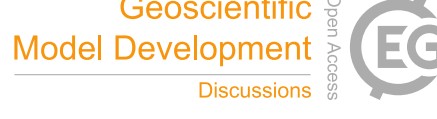

# 1   Introduction

About 10 percent of human settlement is currently and will likely continue to be clustered in regions potentially vulnerable to sea level rise (SLR) (McGranahan et al., 2007), which will arguably result in some of the most devastating impacts of climate change. The polar ice sheets, and their peripheral glaciers, referred to hereafter as "land ice," represent the largest potential

source of SLR in a warming climate through (1) increased meltwater runoff that is not compensated by increasing snowfall (Fettweis et al., 2013; Nöel et al., 2015) and (2) increased outflow (calving and marine melt) (Church et al., 2013). In order to provide credible predictions of SLR to policymakers and stakeholders, scientists need accurate representations of land ice as simulated within Earth systems models.

The scientific community's "best" predictions of ice-sheet mass change rely on process-based models and, increasingly,

model-ensemble predictions from stand-alone ice-sheet models (ISM) (Church et al., 2013). To provide optimal results, these ice sheet models must include accurate representations of ice sheet dynamics, physics, and coupling schemes (e.g., to obtain forcing from other components, like the ocean and atmosphere). As these models are connected to coupled Earth system models (ESM), coupled-model initialization procedures and a quantitative understanding of key model sensitivities and uncertainties (Vizcaíno, 2014) are also required. These challenges are well-recognized: there is an ongoing effort within coupled ESMs to

develop a dynamically active ISM (Lipscomb et al., 2013; Barbi et al., 2014; Ziemen et al., 2014) as well as high-resolution and high-fidelity stand-alone ISMs (Larour et al., 2012; Aschwanden et al., 2016). Furthermore, the ISM community has organized a number of intercomparisons under the umbrella of ISMIP6 that are currently underway (Nowicki et al., 2016; Goelzer et al., 2017) to better understand and compare the distribution of ISM predictions. These could also be used to track model development.

Of course, there are limitations with all models; they are an imperfect representation of actual observed behavior. It is critically important to identify and quantify their most notable biases and discrepancies in order to maximize the impact of model improvements and to increase confidence in model predictions. However, the paucity of observational data in the polar regions, especially data that is specifically relevant to ice sheets themselves, has prevented a comprehensive assessment of ISM and coupled ESM-ISM skill, and key climatological forcings that drive ISM evolution. Although technology has enabled

a significant growth in glacier and ice sheet observations via *in-situ*, airborne, and satellite-based systems in the last decade, this short time frame can only provide current information because ice sheets, more than other aspects of the surface climate system, evolve over much longer time scales. Of course it is not possible to execute experiments in a lab setting where one could adjust parameters to develop an accounting of the sensitivities of the land ice system. While theoretical underpinnings can be used to develop insight toward model sensitivities (refer to Schoof and Hewitt (2013) for a review and summary of

progress), these constructs are also limited by our current understanding of the drivers of ice sheet evolution.

With the aim of facilitating large-scale development and execution of ISM and coupled ESM-ISM experiments, and determining the degree to which they sufficiently represent aspects of the actual Earth system, we present a software package that provides a basic but extensible capability to assess ice sheet models within, and independent of, coupled ESM configurations. The philosophy of verification and validation (V&V), using terminology and standards from Oberkampf and Roy (2010), and



its adoption by the LIVVkit software to verify ice sheet model simulations is presented and discussed in Kennedy et al. (2017). Efforts to validate both ISM and ISM-ESM behavior, including a new capability to compare ESM-derived surface mass balance against recently available observations, is detailed here.

## 2 Target simulations and comparison data

### 2.1 Target simulations for analysis

In order to demonstrate the validation features of LIVVkit, we analyze output from two Greenland ice sheet (GrIS) simulations from (1) a high resolution ISM and (2) a global, fully coupled ESM from which ISM forcing is derived. Both have been presented in the literature and made available to us, which allows us to verify the software for use by others with confidence and complement existing community validation efforts.

For the high-resolution Greenland ISM simulation (1), data were generated from the Community Ice Sheet Model, version 2 (CISM2, Price et al. (2015)), coupled to the Albany-FELIX velocity solver (Tezaur et al., 2015a, b), hereafter referred to as "CISM-A". These simulations are described in detail by Price et al. (2017). This configuration is selected to highlight how LIVVkit could be used to validate a stand-alone simulation that can then be used as an initial condition for follow-on simulations. The simulation used here includes a dynamic ice sheet component forced by a time-varying surface mass balance (SMB) field. It spans years 1991-2013, although these dates should be considered a generic present-day period because the data assimilation techniques used for the initialization used multiple datasets from the last 20 years. The initial state was generated through a multi-step procedure to produce balanced internal temperature and velocity fields, which was then forced by surface mass balance anomalies from the Regional Area Climate Model (RACMO; version 2 (van Angelen et al., 2013)). The goal in creating these simulations was to illustrate the use of the *Cryospheric Model Comparison Tool* (CmCt), an ice sheet model validation tool that focuses primarily on the preprocessing necessary to facilitate the comparison of satellite data to ISM output (Nowicki et al., 2017). That effort is complementary to the validation software presented here, which focuses on comparisons to *in-situ* and remotely-sensed data from other sources.

For the coupled ESM (2), we analyze output from the atmosphere (Neale et al., 2013) and land surface (Lawrence et al., 2012) components within a global, fully-coupled simulation from the Community Earth System Model (CESM1.0)[1]. For this model, the ice sheet surface mass balance (accumulation less runoff and sublimation) is calculated within the snowpack model of the land surface component and then downscaled to the relatively higher-resolution ice sheet grid in order to provide SMB forcing for the ISM component (Rutt et al., 2009), as described in (Lipscomb et al., 2013). Details about the simulation and the SMB are presented in (Vizcaíno et al., 2013) (hereafter V13). Here, we focus on a historical, transient simulation spanning 1850-2005, although we restrict our analysis to 1960-2005 except where noted to avoid issues related to model spin-up. Hereafter, we refer to this simulation as "CESM".

---

[1]http://www.cesm.ucar.edu/models/cesm1.0/cesm/cesm_doc_1_0_6/book1.html



## 2.2 Comparison data specific for ISM

A robust validation that confronts an ISM with a wide variety of independent, observational data is not yet possible. This is, in part, because ISMs currently use much of the most informative observational data for model initialization. As an example to move the community towards validation, we apply LIVVKit to generate and present model-data comparisons using as many
available datasets as possible and including those that are used for model initialization.

For validation of surface velocity and ice thickness, the data available for comparison are initialization data. These were obtained using a PDE-constrained optimization procedure (Perego et al., 2014). We include these in LIVVkit with the expectation that independent data will replace this data within the same workflow going forward. LIVVkit makes use of a Github repository of scripts and procedures (`jhkennedy/cism-data`; Kennedy, 2017) that procures the surface velocity and
thickness datasets from a host of publicly available sources. The workflow processes this dataset to create a number of fields of variables in an expected format, with all the masking and projections necessary for postprocessing in LIVVkit (explained in section 3). The velocity fields used for comparison originate from the NASA *Making Earth System Data Records for Use in Research Environments* (MEaSUREs) program, which provides annual ice-sheet-wide velocity maps for Greenland, derived using Interferometric Synthetic Aperture Radar (InSAR) data from the RADARSAT-1 satellite (Joughin et al., 2010a, b). The
dataset currently contains ice velocity data for the winter of 2000-2001 and 2005-2006, 2006-2007, and 2007-2008 acquired from RADARSAT-1 InSAR data from the Alaska Satellite Facility (ASF), and a 2008-2009 mosaic derived from the Advanced Land Observation Satellite (ALOS) and TerraSAR-X data. For bed topography and ice thickness, we use the Greenland Ice Mapping Project Digital Elevation model (GIMP DEM; Howat et al., 2014) for topography of the ice free areas, and the Morlighem et al. (2014) bed topography and ice thickness estimates, which are derived from ice surface elevation data, airborne
radar soundings from Operation IceBridge, and the GIMP DEM (as a reference surface). It is straightforward to update and/or augment these same data with new years and locations as they become available (e.g., Morlighem et al., 2017).

## 2.3 Comparison data specific for ESM

In the case of coupled ESM validation for ISM development, modelers will benefit from more focused and quantitative evaluation in the vicinity of the GrIS, as compared to the global and regional model validation typically provided with the release of
components that comprise ESM. Therefore, LIVVKit provides validation of the atmosphere and land surface components in ESM specifically over the GrIS region for key variables that affect it. Validation of the atmosphere and land surface is still limited by the availability of observed data over the GrIS (and Antarctic ice sheet), whether it is used directly or within reanalysis products. As with the ice sheet data, additional regional in-situ and remotely sensed data will improve LIVVkit and is a target for further development.
From LIVVkit, the examples presented in section 4.2 include cloud fractions from an ESM, which are compared to the International Satellite Cloud Climatology Project (ISCCP) (Rossow and Schiffer, 1999) and the combined CLOUDSAT Radar and CALIOP Lidar datasets (hereafter CLOUDSAT) (Kay and Gettelman, 2009) reanalysis products. ISCCP resolves the diurnal cycle, within which monthly averages are created, and provides the longest available time record. CLOUDSAT covers





only a short time record, however the detection techniques are considered superior for the Arctic. The interested reader is referred to the Climate Data Guide (Pincus, R. and NCAR Research Staff (Eds.)., 2016) for more details about the attributes and limitations of these datasets.

## 2.4 Surface Mass balance comparison data

Given the dependence of ice sheet evolution on surface mass balance, LIVVkit tracks SMB for both coupled ESM and stand-alone ISM, even if it is provided as a forcing for a simulation for the ISM. For an ice sheet wide view of SMB behavior, LIVVkit presents metrics relative to the most recently available version of RACMO, version 2.3 (RACMO2.3), the characteristics of which are summarized in Nöel et al. (2015). The configuration of RACMO 2.3 has been specifically designed and validated for its fidelity in capturing the extended Greenland region. As with initialization data, model to model comparisons of SMB

data do not provide independent validation. This is especially true when applied to a standalone ISM that has been forced with model SMB, as is the case for CISM-A, which has been forced with RACMO2.0 (presented in section 4.1). We include these comparisons within LIVVkit because it is useful when (1) presented against ESM SMB and (2) to have available when monitoring other aspects of ice sheet evolution to attribute sources of bias. RACMO2.3 data is provided at approximately 11 km horizontal resolution with 40 vertical layers, and includes the GrIS and other nearby areas. LIVVkit is currently configured

to compare years 1980-1999 of the RACMO2.3 simulation because it was forced with the more recent ERA-Interim reanalysis data (Stark et al., 2007; Dee, D.P. et al., 2011) from 1979-2014 and many of the model development runs for Earth system models cover the period spanning 1979-2000 as part of the Atmospheric Model Intercomparsion Protocol (AMIP-II) (Gleckler, 2004).

To provide independent validation of an ESM SMB field, LIVVKit compares *in-situ* SMB values using 623 core/pit/stake

measurements from a diversity of sources. Many of these were included in Vizcaíno et al. (2013, Fig. 6), but have been updated in LIVVkit with new data and selection criteria. For a station to be included, it must contain a record of location, elevation, and observation start/end dates. Here, we use 387 locations out of a possible 450 from a collection of accumulation zone (net specific SMB > 0) estimates compiled by Cogley (2004) using data from Ohmura and Reeh (1991), Ohmura et al. (1999), and Bales et al. (2001). In addition to the stations included from Bales et al. (2001), 38 additional ice cores from PARCA

(Bales et al., 2009) have been added, with 6 of those as replacements to the earlier 2001 study. The PARCA compilation also includes reanalyses of time series from 20 coastal weather stations to estimate SMB, however only estimates from ice cores and snow pits are currently included in LIVVkit. We also include data from PROMICE, a compilation of glacier explorations and *in-situ* measurements taken since the 1960s from the GrIS ablation zone (SMB<0) (Machguth et al., 2016). From 790 original locations, all but 198 stations were excluded based on the following criteria:

– Unknown elevation or start or end date of observation period,

– Observation period was less than 95% of a year (i.e. seasonal data),

– Accumulation data were derived using a methodology other than pit/core measurements (e.g., weather stations or surface lowering relative to ramp road).





For all stations, temporal data were aggregated and treated as typical climatology, regardless of the record length. If annual SMB estimates were supplied for multiple years, LIVVkit averages over all years to provide a single annual value for each location. Because station selection is subject to various adjustments based on the type of validation to be performed, we choose this selection criteria to facilitate a general, starting-point comparison.

Model results are also compared to accumulation estimates derived from 25 NASA Operation IceBridge radar airborne flights from the 2013 and 2014 seasons as detailed in Lewis et al. (2017) and the associated supplementary data. The measurements capture internal reflecting horizons in the top few hundred meters of the ice sheet, which can be used to estimate the historical SMB record spanning the past three centuries. The IceBridge data from Lewis et al. (2017) are provided as raw estimates seasonally for a given lat/lon coordinate. Because the temporal record varies for each site, LIVVkit calculates annually averaged

SMB at each location.

In order to provide basin scale estimates of SMB, we use the Zwally et al. (2012) drainage basin delineations as visualized by color in Figure 1 (colors) and with numbering conventions preserved. This figure also illustrates the location and extent of basin-wide SMB data, including pit/core locations (filled shapes) and IceBridge altimetry transects (white lines). Accumulation zone SMB estimates from Cogley (2004) and Bales et al. (2009) PARCA cores are shown as blue circles, while ablation zone

PROMICE data (Machguth et al., 2016) are shown as yellow triangles. The data points are sized by the length of their temporal record, with larger markers indicating that estimates were averaged from a greater number of annual SMB values.

Processing the IceBridge data from Lewis et al. (2017), which are provided as raw estimates seasonally for a given latitude/longitude coordinate, is a two-step process in LIVVkit. Each model cell is assigned to a Z12 basin. It does this by converting the Z12 drainage basin outlines into polygons, and then decides which polygon contains which model cell centers. If a model

cell center is not within any of the basin polygons, it is assigned basin '0'. Correspondingly, each IceBridge measurement at a lat/lon location is averaged over seasons to obtain an annual SMB value. Some locations have older SMB records so the average annual SMB value used for comparison to the model is the mean over all available temporal records. Then a kd-tree/nearest neighbor method is used to find the model cell closest to the observed lat/lon measurement. Thus, the Z12 basin at that location will be the same one as the corresponding model cell, found in the first step.

The present strategy to include both observational and model comparison data was chosen to maximize (1) clarity (the scientist understands the limitations of the information); (2) reproducibility (automation within LIVVkit); and (3) extensibility (users can add additional data for their own comparisons with minimal local adaption of the code). For provenance, changes to LIVVkit's inclusion of new data are controlled through releases so users can be certain of the data they are comparing for a given tag of the data repository. LIVVkit can be altered at will once the code is forked or downloaded locally to suit the user's

needs. The steps to process comparison data is a separate step explained in section 3, but it is automated and fully documented. A discussion about potential new candidate data to add to the current collection is provided in section 5. Additional details and references to the comparison data can be found at the github site (Kennedy et al., 2016).





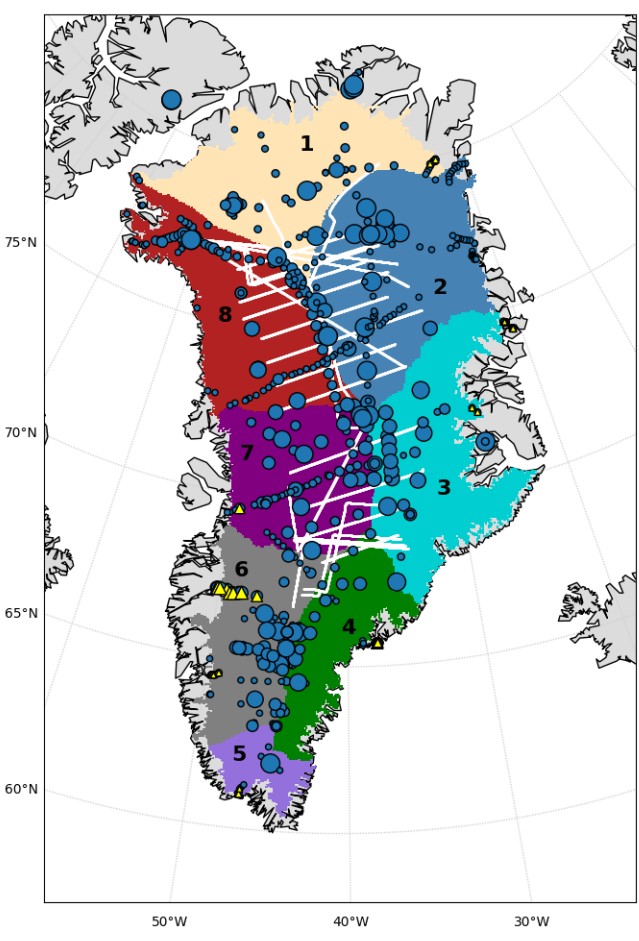

**Figure 1.** Greenland ice sheet drainage basin delineations (colored and numbered regions) as in Zwally et al. (2012), the location and temporal extent of the pit/core locations (filled circles and triangles), and the altimetry transects (white lines) used in LIVVkit SMB analyses. The basin numbers follow Zwally's designation and the colors correspond to those used in histograms and scatter-plots throughout. Pit/core data points are sized by the length of their temporal record, with larger points indicating annual estimates were taken from a greater number of yearly SMB values.





## 3 Software infrastructure for validation

LIVVkit is a Python-based, open source software package designed for verification and validation of ice sheet and Earth system models. LIVVkit operates on model output, viewing the model as a black box. This strategy provides flexibility in analyzing many different models, assuming some basic conventions are followed. LIVVkit is installable through Anaconda/Miniconda, PiPy, or github, and provides the `livv` command line interface (CLI), which is used to execute the analyses and output the results to a portable website on the user's local machine. The details of the design philosophy, construction, and the verification components of LIVVkit are described in Kennedy et al. (2017), so only the validation components of LIVVkit are presented here.

There are two major steps to validate a model simulation (or set of simulations), assuming that the model output is already accessible on the analysis machine, (1) postprocessing the raw model output data and (2) analysis/visualization of the prepared fields. For models run at Leadership Class Computing Facilities (LCF), output is typically located on cross-mounted file systems that allow access from both computing and processing nodes. First, the output needs to be organized to facilitate analysis. Typically, model output is organized for optimal writing, not scientific analysis. Postprocessing also includes data preparation such as interpolation to a common projection, treating missing points consistently, and preparing the metadata for each field. The comparison datasets require similar gathering and processing steps ahead of the model analyses, if they are being used for the first time.

LIVVkit provides a set of single execution, task-parallel postprocessing (bash) scripts designed for automated postprocessing of model output at LCFs, so that the data is prepared for scientific analysis through (and outside of) LIVVkit. These scripts can be adapted for other systems. Earth system models are comprised of multiple component models (e.g., land, ocean, atmosphere, sea ice, land ice) that may either be active or provided as a data model, depending on the configuration selected. LIVVkit currently targets the atmosphere, land surface, and ice sheets if they are active, creating a new directory set by the user that contains subdirectories for each component and all their final data products, including their incumbent metadata and associated masking and mapping files. This enables full reproducibility and structure for additional analysis within or independent of LIVVkit.

The LIVVkit postprocessing scripts use a combination of Python and netCDF Operators (NCO) (Zender, 2008), an open source collection of programs that operate on gridded scientific data. LIVVkit uses `ncclimo` commands within versions NCO/4.6.9 and later to facilitate the extraction of monthly, seasonal, and annual means (and optionally, to regrid the data) as well as its baseline commands to average, sum, produce weighted and masked data, and other operations typically used in geoscientific analyses. NCO addresses provenance and transparency by appending the specific details of operations within the metadata of datasets it processes and it performs task parallelism where applicable.

For stand alone ice sheet model output (i.e., where climate forcing fields such as surface temperature and surface mass balance are *not* provided from coupling to a climate model), LIVVkit processes the SMB data used to force the model, thickness, three-dimensional temperature and horizontal velocity, and surface elevation, and it assumes that the data is located within a





single file. From the velocity components, it will create the velocity norms. When complete, the postprocessed data directory contains:

- multi-year monthly, seasonal, and annual climatologies (time averages) over the selected period for all variables

- time series of all variables and their annually and area-weighted averages over the length of output for selected variables

5     - annual and seasonal ice sheet mask area-weighted, and GrIS masked averages for selected variables over the selected period of analysis.

For GrIS masked data, values are averaged over a region defined by a minimum ice thickness (0.001 m default, used for this analysis). For velocity, areal averages are computed over the cells where they are defined.

When processing large scale coupled Earth system model simulations, to which an active ice sheet may or may not be 10   coupled, LIVVkit targets radiation, thermodynamics, hydrological, and dynamical variables that influence ice sheet evolution. This process is more extensible and robust than for stand-alone ice sheet model processing, because coupled models follow a specific Climate and Forecast (CF) metadata convention. For the coupled model, the completed postprocessed data directory contains:

- multi-year monthly, seasonal, and annual climatologies over the selected period for all variables

15     - time series of annually and area-weighted averages and GrIS masked for selected variables, each in a separate file, over the length of output

- annual and seasonal area-weighted and GrIS masked averages for selected variables over the selected period of analysis

- annualized daily averages for selected variables over the full length of the simulation.

within subdirectories for each component.

20   Processing high resolution data requires computing nodes with high memory. These "fat" nodes can be accessed using batch nodes at LCF, and LIVVkit's processing scripts handle the batch queue submission. Likewise, LIVVkit provides a LIVVkit Extensions (lex) git repository, with large file support, that contains already processed observational and comparison data for use with LIVVkit. This repository is hosted at Oak Ridge National Laboratory (ORNL) and is available to the public (Kennedy and Evans, 2018). This repository is broken into subdirectories containing different validation analyses where the processed 25   data, a detailed description of the data including provenance, a LIVVkit configuration file, and a Bibtex file are provided. For each validation analysis, LIVVkit uses the data description to provide some context to the analysis and requests that the original data authors are cited in any publication which utilizes the analyses for testing, development, or scientific analysis. The appropriate citations are detailed in the provided Bibtex file and listed on the output website. The lex repository is already sizable, as it contains some large (gigabyte scale) observations datasets, and is expected to grow to unwieldy proportions as more 30   observational data is included and older datasets are updated. In order to handle the projected size of this repository, LIVVkit details the commands necessary to only clone the analysis files without the data files (e.g., the description, configuration, and

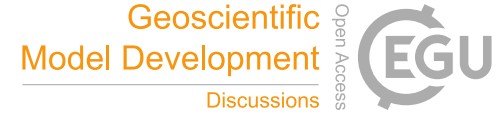



Bibtex files) in order to see which analyses are available, and then pull down just the latest version of the required data to run the analyses.

It is important to acknowledge that there are numerous intellectual property concerns with hosting and redistributing datasets. To satisfy these concerns, lex maintains a public-private development cycle where analyses are developed in a private git repository (also hosted at ORNL) and only released to the public repository once dataset author/maintainer permission is provided (LIVVkit itself is publicly developed). For analyses that rely on datasets where permission cannot be granted, LIVVkit will provide a description of how to acquire the data and scripts to process the data for analysis. Currently, the LIVVkit development team is pursuing partnerships with data providers like NASA and NSIDC to reduce or eliminate the time to get new analyses into the public lex repository. We note that the lack of standards/adoption and/or metadata conventions, e.g., CF-Conventions, has been the biggest challenge for the rapid inclusion of a dataset in lex, rather than intellectual property concerns.

Once the postprocessing is complete, the prepared model and observational/comparison data are analyzed over a suite of time and space slices and are provided to the users in a combination of easily digestible text, tables, and figures. A validation analysis is initiated by pointing the LIVVkit execute command, `livv`, to a JSON[2] configuration file that specifies the analysis details:

```
livv --validate /path/to/analysis.json -o $OUTDIR -s
```

Within the JSON configuration file, the user provides the type of model run and the location of the output and comparison data. The user may also specify the location where the output data and website will be placed when complete (via the `-out-dir` or `-o` option) and launch a simple HTTP server to host the website (via the `-serve` or `-s` option). When the HTTP server is launched, a direct hyper-link is printed on the command line to either the analyses that were just executed, or a previous analysis selected via the CLI. For example, all the figures and analyses presented here can be reproduced by executing the following command from the top directory of lex.

```
livv --validate smb/smb_icecores.json \
                energy/energy_cesm.json \
                clouds/clouds_cesm.json \
                dynamics/dynamics_cisma.json \
                -o vv_evans2018 -s
```

While any particular analysis performed by LIVVkit can be quite sophisticated, creating a new analysis in LIVVkit/lex is intended to be straightforward for the user. The processes of building an extension is detailed in the LIVVkit documentation. A template validation analysis is provided within the software that details the minimal python code needed to run an analysis script, along with a minimal template JSON configuration file. A user can place any external information needed by their analysis in the configuration file (e.g., paths to data, years to analyze, method switches). For sophisticated analyses, LIVVkit is also able to import analysis packages developed with a standard python package structure as long as a configuration file and

[2]JavaScript Object Notation is a lightweight, language-independent tool that both machines and humans can parse.



| Variable | Units | Model Areal average | Input Areal Average |
|---|---|---|---|
| SMB | kg m$^{-2}$ a$^{-1}$ | 259.0 | 255.9 |
| Top layer temperature | $^{o}$C | -2.3 | N/A |
| U velocity | m a$^{-1}$ | -8.3 | -14.5 |
| V velocity | m a$^{-1}$ | -3.1 | -0.8 |
| Velocity norm | m a$^{-1}$ | 54.3 | 62.7 |
| Ice thickness | m | 2179. | 1781. |

**Table 1.** Area weighted and GrIS masked annual averages for variables from CISM-Albany model output. "Input Areal Average" refers to the processed datasets from observations as described in section 2 that are used for the initial state of the model before spin up.

a LIVVkit entry point are provided. The lex repository also functions as a set of examples and contains both the basic script and package style analyses.

## 4 Presentation and Visualization

Targeting stand-alone ice sheet and coupled Earth system model output within LIVVkit provides two common use cases of
model evaluation of ice sheet behavior and drivers. A subset of these analyses are presented below, with a special focus on a basin-wide analysis of the surface mass balance using more recent observational data presented in section 2.

### 4.1 Stand-alone land ice model analysis

Using GrIS simulation data, processed as described in section 3, LIVVkit produces a website with metrics and plots for validation. As an overall check of model behavior and to identify potential large-scales biases, there is a table of annualized
and areal average values over the selected time record for relevant variables, which includes the surface mass balance (SMB) forcing, dimensional velocity (e.g. zonal (U) and meridional (V) velocity components, respectively), and velocity norm. Table 1 displays a similar version of the table produced by LIVVkit as applied to the CISM-A simulation described in section 2.

LIVVkit displays a suite of two-dimensional contour plots of climatological averages over the selected time record. Figure 2 presents the surface mass balance from CISM-A, RACMO2.3, and their difference. Because the CISM-A simulation was forced
with RACMO2.0 and not RACMO2.3 data, there are differences between CISM-A and RACMO2.3 that mirror the differences between the RACMO versions, in particular the Southwestern and Northern melt regions and over strong accumulation regions (Nöel et al., 2015). Although for this example of SMB within CISM-A, the comparison is somewhat contrived to illustrate capability, SMB forcings can be monitored along with output variables to track their impact on the simulation. Scatter plots comparing the same data are also displayed, whereby areas with close correspondence between the two datasets are clustered
along the red identity line. As applied to CISM-A versus RACMO2.3 for the SMB (Figure 3), generally the differences between the model and RACMO2.3 surface mass balance values are small (as expected given the origin of the SMB forcing data).



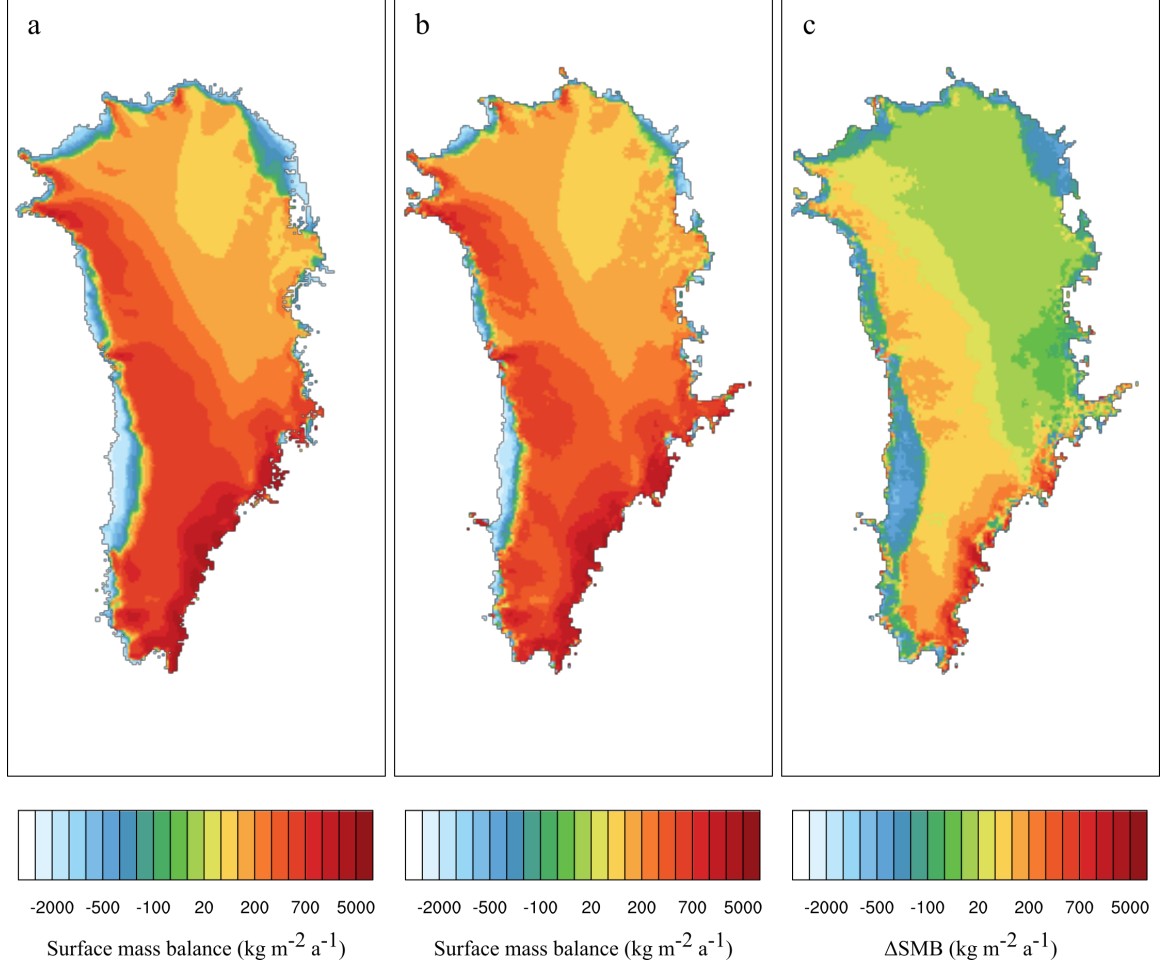

**Figure 2.** Surface mass balance contours for (a) CISM-A forced with RACMO2.0, (b) RACMO2.3, and (c) CISM-A minus RACMO2.3, with CISM-A interpolated to the coarser RACMO2.3 grid.

LIVVKit also presents contour figures for the GrIS climatology of top level temperature (without comparison data, not shown), and velocity $L_2$ norm for a stand-alone model. The norms are created for the entire GrIS and several regions of interest, which are those that highlight the Jacobshavn, Zachariae, and Petermann glaciers. Figure 4 is focused around drainage basins 7 and 8 (see Fig. 1 for reference) to include the Jakobshavn glacier.

## 4.2 Coupled model analysis

Land ice behavior is critically dependent on various forcings from the atmosphere and land surface, so analyses of these fields within a coupled model comprise a significant portion of LIVVkit validation. LIVVkit also targets the land ice component within a coupled simulation using the same analysis procedures described in section 4.1 for a stand-alone land ice model, if



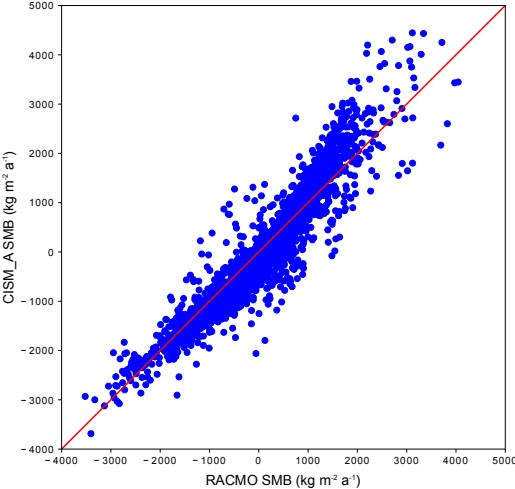

**Figure 3.** Scatter plot of all grid points of SMB for RACMO2.3 versus CISM-A interpolated to the RACMO2.3 grid

that component is active. As with the stand-alone analysis, LIVVkit presents area averaged, climatological fields in tabular form for a "quick look" overview applied to the CESM simulation and RACMO2.3 comparison data, as shown in table 2. When applying the GrIS mask for averaging, the variables are multiplied by the percent of grid cell that is land ice, which is constant for the CESM simulation. Within the land model component, the downwelling and net shortwave ($SW_d$ and $SW_{net}$)

and longwave ($LW_d$ and $LW_{net}$), respectively, the net surface radiation, $R_{net}$, the turbulent sensible and latent heat fluxes ($SHF$ and $LHF$), and the SMB are all summarized. From the atmosphere component, the 2 meter air temperature (T2m) is presented. Here, LIVVKit produces identical results for CESM output as in V13 except for the SMB, because LIVVkit uses monthly averaged values of the ice growth/melt (QICE) field.

  The variables summarized in Table 2 are also presented as contour plots over GrIS to highlight regional differences from

10 RACMO2.3. Annualized and/or seasonal fields (as appropriate) from land surface and atmosphere are provided for the model, RACMO2.3, and the difference, where RACMO2.3 is interpolated to the CESM grid. The summer $LW_{net}$ in CESM is presented in Fig. 5 as an example, and shows that although the model is able to capture the general spatial variation and minimum values near the NE coast, there are overly large values (less heat loss) near the center of the land ice. Contour and scatter plots of climatological atmospheric 2 meter height temperature are also provided for polar summer (JJA) and winter (DJF) in

LIVVkit; a contour plot for JJA is shown in Fig. 6. The near surface temperature is critical to track in a coupled model because it drives the processes of surface melt and refreezing.

  The representation of clouds over GrIS in the coupled model is also evaluated in LIVVkit because they contribute to the surface energy budget and related processes such as surface melting and refreezing (Van Tricht et al., 2016). The comparisons provided by LIVVkit include the annual cycle of monthly averaged low, high, and total clouds for both the model and ISCCP



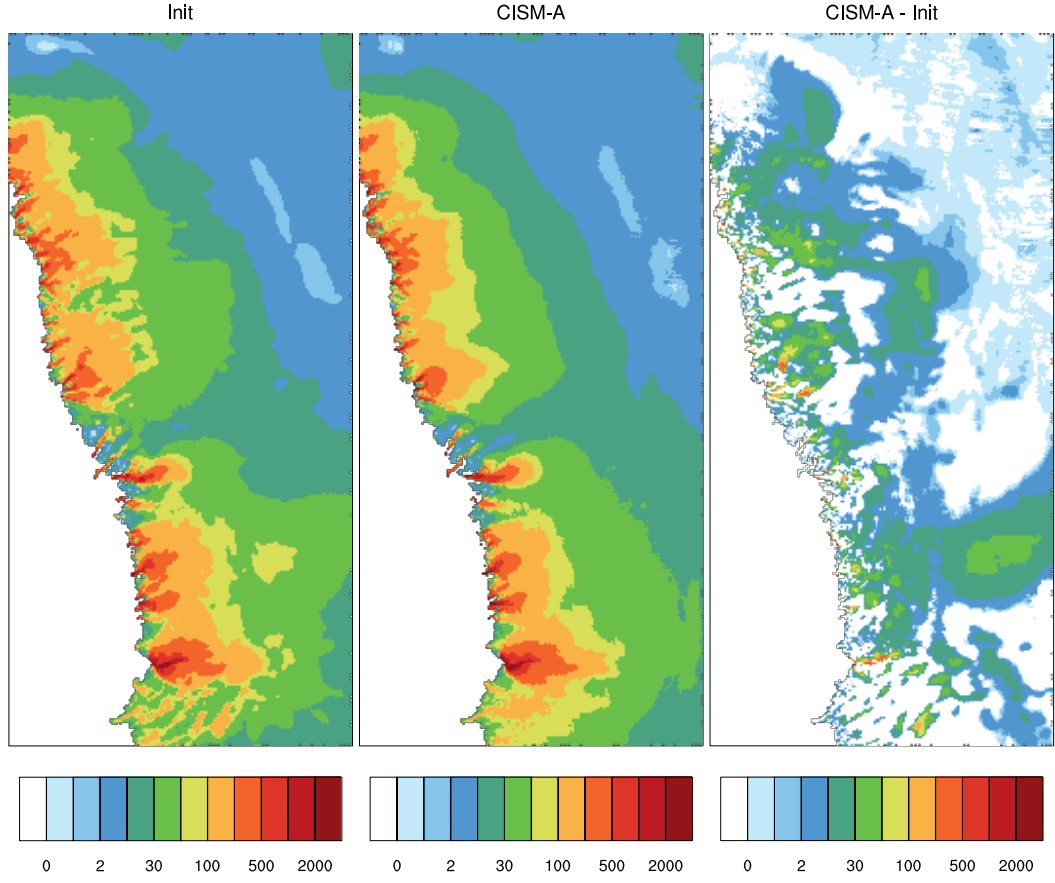

**Figure 4.** The norm of the surface velocity field (m/s, given by the color bars) from initialization (Joughin et al., 2010a, b) (left), CISM-A (middle), and the difference (left) over drainage basins 7 and 8 as indicated by figure 1.

and CLOUDSAT observationally derived datasets and contour plots of annual and seasonal averages of low, high and total cloud amount. For the annual cycles, the data are area-averaged over the GrIS region for each month of the climatology. Several examples of the cloud analysis as applied to CESM are displayed in Figures 7 and 8. The ISCCP and CLOUDSAT monthly averages over GrIS in Fig. 7 exhibit a seasonal cycle, which is consistent with findings over the entire Arctic (Chernokulsky and Mokhov, 2012). However, the CESM produces total clouds that are considerably too few and capture no summer minimum, also consistent with noted CESM biases observed for the whole Arctic region (Barton et al., 2012). This bias is not universal for all cloud levels; Fig. 8 shows that the annual values of low clouds more closely match CLOUDSAT than ISCCP (although the seasonal cycle is opposite from observations, with a summer maximum, not shown). These plots indicate the need for a deeper investigation. Recent efforts to understand the limitations of both models and observed and reanalysis datasets in representing clouds over the Arctic (e.g.Chernokulsky and Mokhov (2012); Lenaerts et al. (2012)), and focused over Greenland (Hofer et al., 2017), provide a good starting point.



| Data | CESM | RACMO2.3 |
|------|------|----------|
| $SW_d$ | 268 | 301 |
| $SW_{net}$ | 61 | 58 |
| $LW_d$ | 235 | 224 |
| $LW_{net}$ | -46 | -52 |
| $R_{net}$ | 15 | 5.5 |
| SHF | 7 | 8 |
| LHF | -8 | -6 |
| SMB | 209 | 238 |
| T2m | -20.6 | -21.9 |

**Table 2.** Area weighted and GrIS masked climatologies for a host of key land surface variables from CESM's land surface and atmosphere components for summer (JJA), except SMB, which is an annual climatology, compared to RACMO2.3 values. All variables are in W m$^{-2}$ except SMB (kg m$^{-2}$ a$^{-1}$) and T2m ($^o$C).

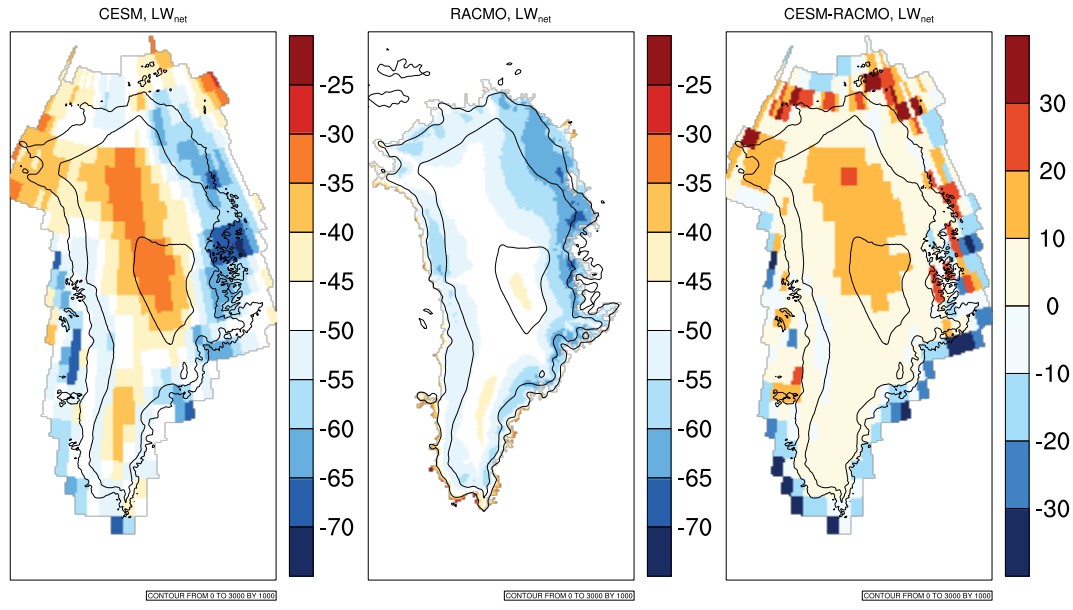

**Figure 5.** Summer net longwave radiation (W m$^{-2}$) for CESM (left), RACMO2.3 (middle), and the difference (right). The block solid lines denote elevation at 0, 1000, 2000, and 3000 km levels each grid. The CESM elevation (left and right) uses the 5 km downscaled values.



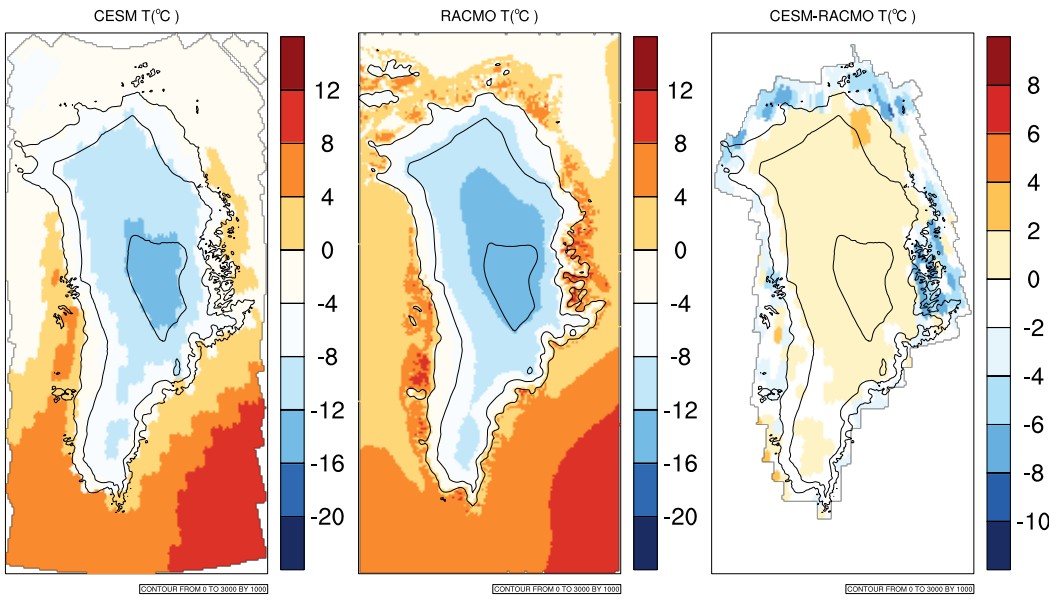

**Figure 6.** Averages for the JJA season of surface temperature over the GrIS for CESM (left), RACMO2.3 (middle), and CESM-RACMO2.3 (right). Elevation contours are as in Fig. 5.

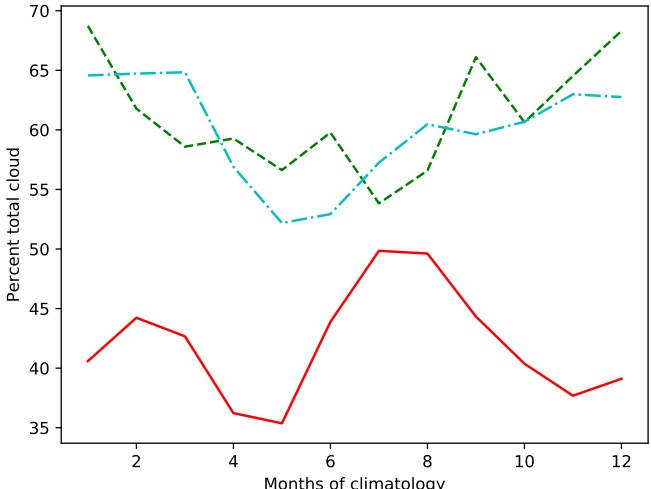

**Figure 7.** Climatological monthly averages of total clouds over the GrIS region for CESM (red), ISCCP (Rossow and Schiffer, 1999) (green), and CLOUDSAT (Kay and Gettelman, 2009) (cyan).





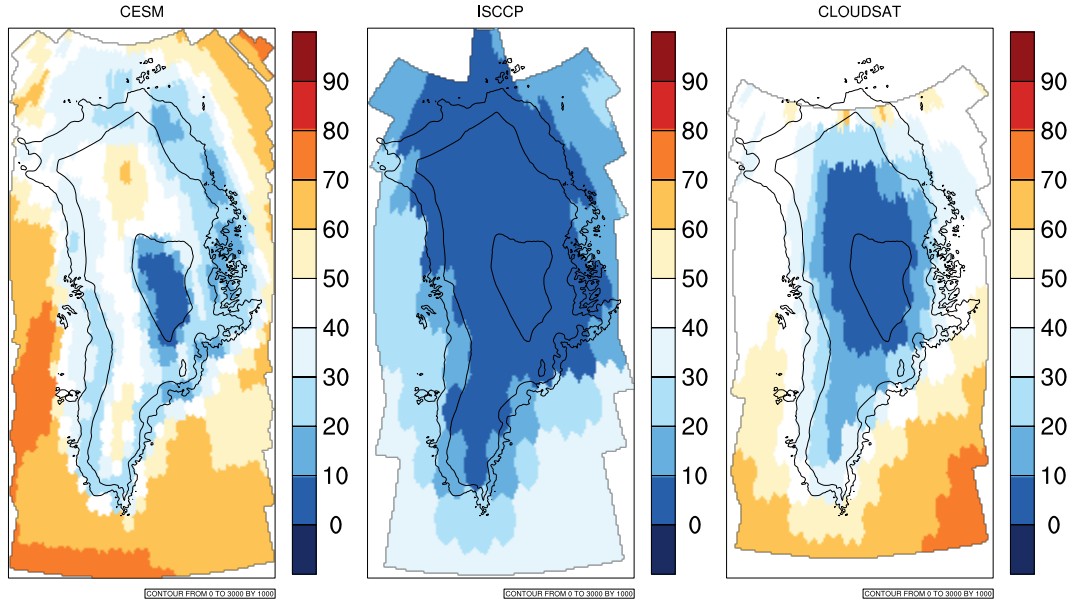

**Figure 8.** Climatological annual average of low clouds (%) over the GrIS for CESM (left), ISCCP (Rossow and Schiffer, 1999) (middle), and CLOUDSAT (Kay and Gettelman, 2009) (right). The downscaled CESM values are used for all elevation contours.

Similar to Fig. 2 for CISM-A, LIVVkit produces a contour plot of ice sheet wide SMB compared to RACMO2.3 (not shown). For the coupled simulation, a times series for the area averaged values over GrIS covering the entire range of a simulation is provided by LIVVkit, as in Fig. 9 for CESM. The values are calculated with the same methodology as with the SMB in Table 2, except for the time averaging. To break down climatological behavior of the time series and quantify the variability, a box plot of the SMB time series is provided, as shown in Fig. 10 for CESM and RACMO2.3. In this plot, the rectangle spans the 25% quartile to the 75% quartile (the interquartile range, or IQR) of the time series with the median shown as the red line in the rectangle. The two whiskers represent $1.5 \times IQR$ above the 75% quartile and below the 25% quartile, respectively. The diamonds outside the whiskers are suspected outliers. In FIg. 9, the extent of the whiskers are similar for CESM and RACMO23, so the model data has similar variability compared to RACMO2.3. However the slightly smaller and lower IQR box for the RACMO2.3 data indicates that its data are slightly less variable and skewed low. Given the smaller dataset size of RACMO2.3, this result is not surprising.

## 4.3 Basin scale SMB analysis

The increasing availability of SMB observational data outlined in section 2 presents an opportunity to delve into the quality of simulated SMB. With this in mind, LIVVKit provides analyses of the SMB by basin and elevation using these data, applied here to the CESM SMB values that have been downscaled to 5 km within the land ice component. A LIVVKit contour plot of





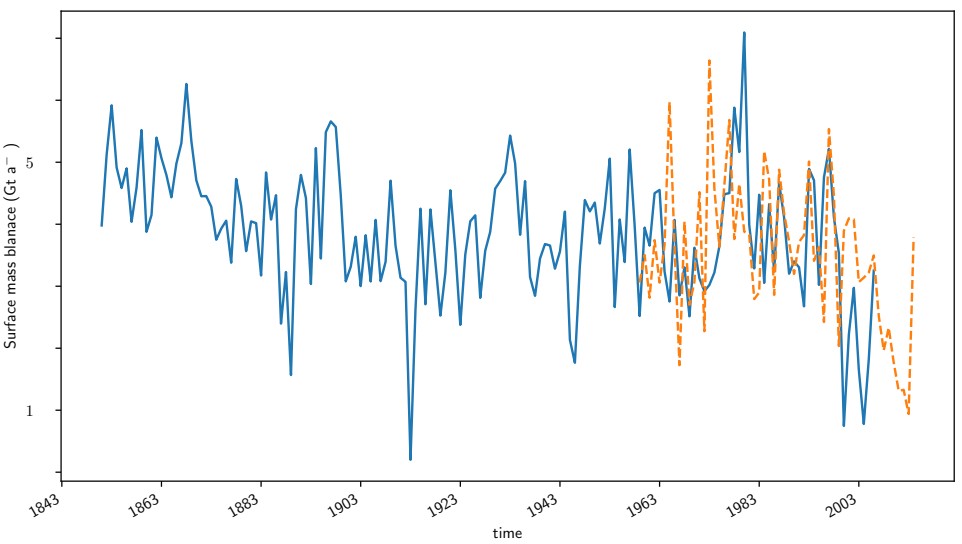

**Figure 9.** Times series of the (blue) CESM surface mass balance for the entire simulation, 1851-2006, compared to the (orange) RACMO 2.3 surface mass balance.

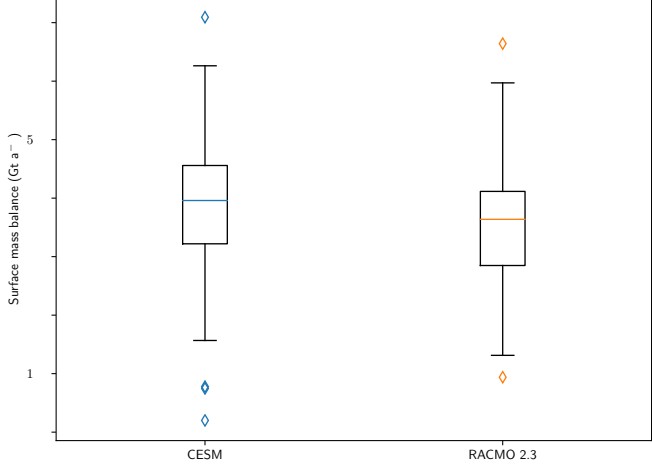

**Figure 10.** Box plot of (left) CESM annual surface mass balance for the entire simulation, 1851–2006, compared to the (right) RACMO 2.3 annual surface mass balance for 1961–2013. The diamonds represent extreme values within the series as explained in the text.





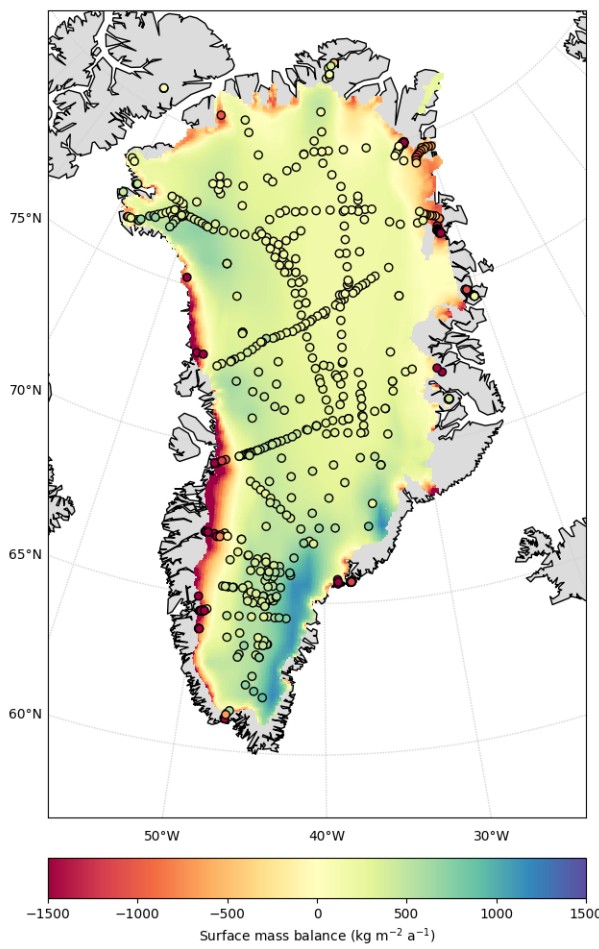

**Figure 11.** Filled contours of the annual SMB of the Greenland land ice as modeled by CESM, with pit/core field estimates overlaid as filled circles. Data were compiled as detailed in section 2 from both ablation and accumulation zones.

the CESM SMB, overlain by circles representing the observed data locations (as in Fig. 1) is shown in Fig. 11. This figure is similar to Figure 7 of V13, but includes more recently available data. The colors within the circles represent SMB estimates based on snow pit and/or firn/ice core studies.

To provide more quantitative comparisons, this data is also presented as a histogram of differences between modeled and observed annual surface mass balance values at pit/core locations where data exists. It is shown for the entire GrIS in Fig. 12





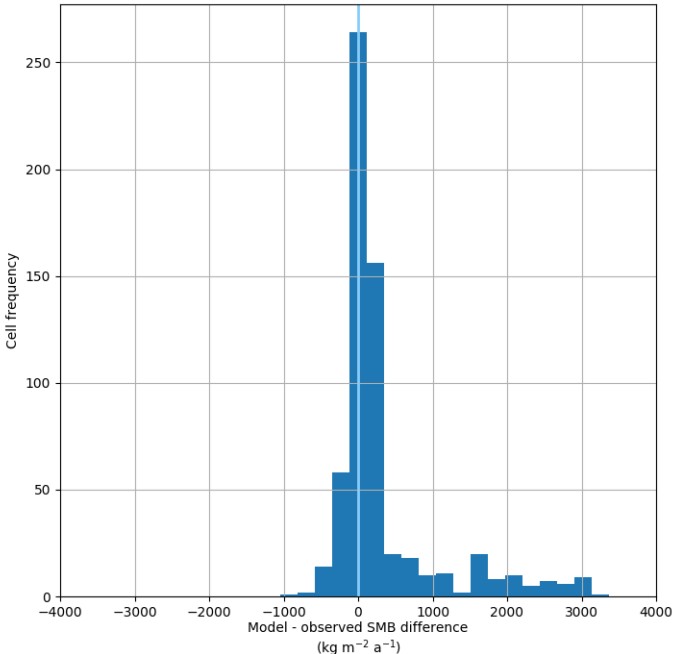

**Figure 12.** Histogram of differences between modeled and observed annual SMB, at all collected pit/core locations as detailed in section 2 from ablation and accumulation zones. The light blue line highlights 0 difference between model and observed; values above (below) this line indicate that the model overestimates (underestimates) SMB in comparison to the observations.

and for each of the eight basins delineated in Fig. 1, in Fig. 13. Observational data for Figures 12 and 13 were compiled and processed by LIVVkit from the pit/core data as described in section 2. The vertical light blue line denotes the 0 difference line for model vs. observed data; values above (below) this line indicate that the model overestimates (underestimates) SMB in comparison to altimetry observations. High frequencies near zero imply greater model agreement. Note that because this plot compares nearest neighbor values without any spatial interpolation, one coarse model cell may be compared to multiple (if not many) observed SMB estimates, because the data collection sites are often located in clusters. The correspondence of model to observational data varies significantly by basin, and shows better agreement in the central latitude regions as compared to the southern and northern regions. V13 showed lower values of SMB over region 4 for CESM relative to RACMO2.0, but in fact CESM is rather close to observations in this region. Consistent with this, RACMO version 2.3 exhibits a significant decrease in precipitation (Nöel et al., 2015), apparently bringing its SMB closer to observations. LIVVkit also provides plots similar to those in Figures 12 and 13, but with comparison to the IceBridge data.

Scatter plots comparing CESM to the SMB estimates from pit/core data and IceBridge areal estimates, separating accumulation and ablation zone and colored by basin, are also provided by LIVVkit. Figure 14, which shows pit/core data, demonstrates





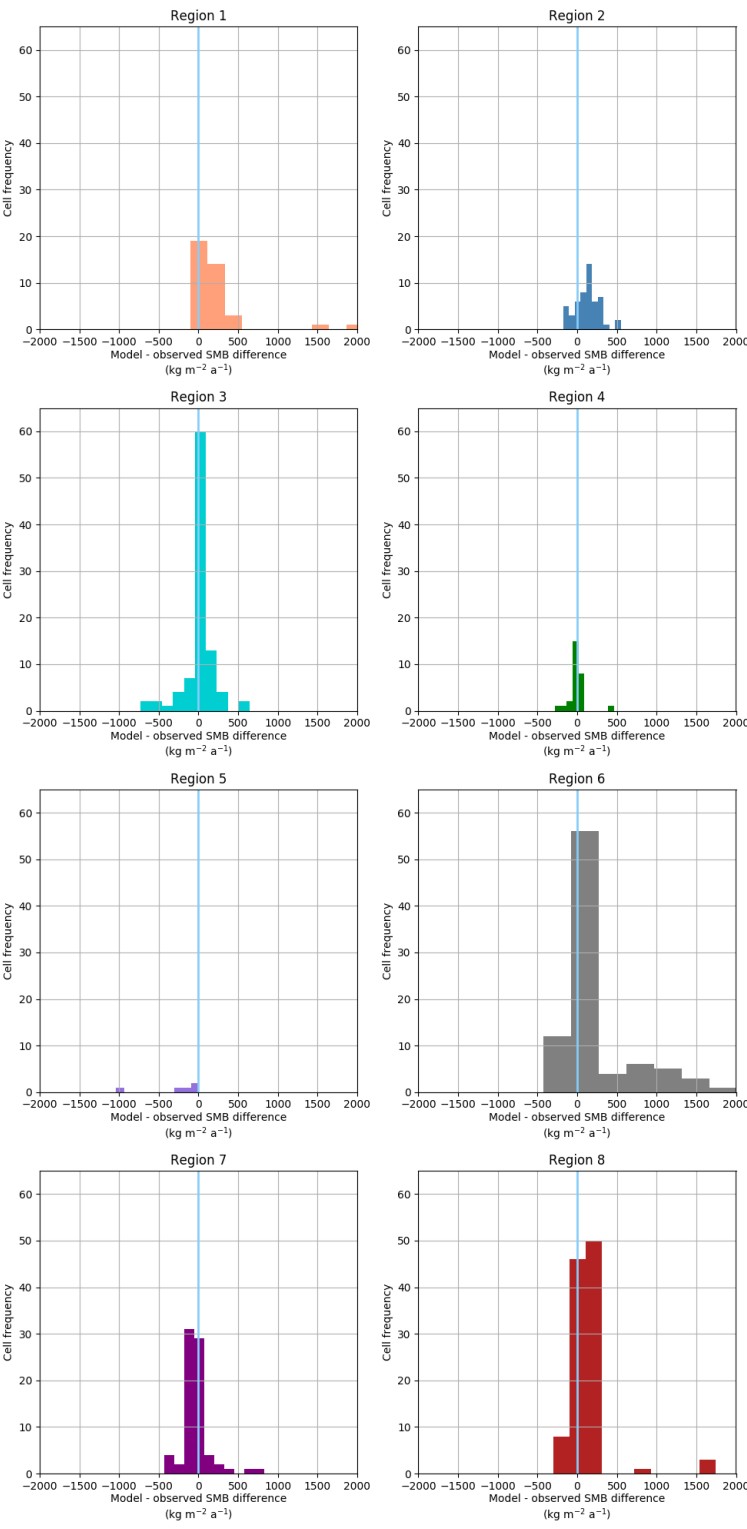

**Figure 13.** As in Fig. 12, but broken down into basins as marked in Fig. 1. Axes have been normalized for the eight drainage histograms to highlight differences between the number of comparison points per basin.





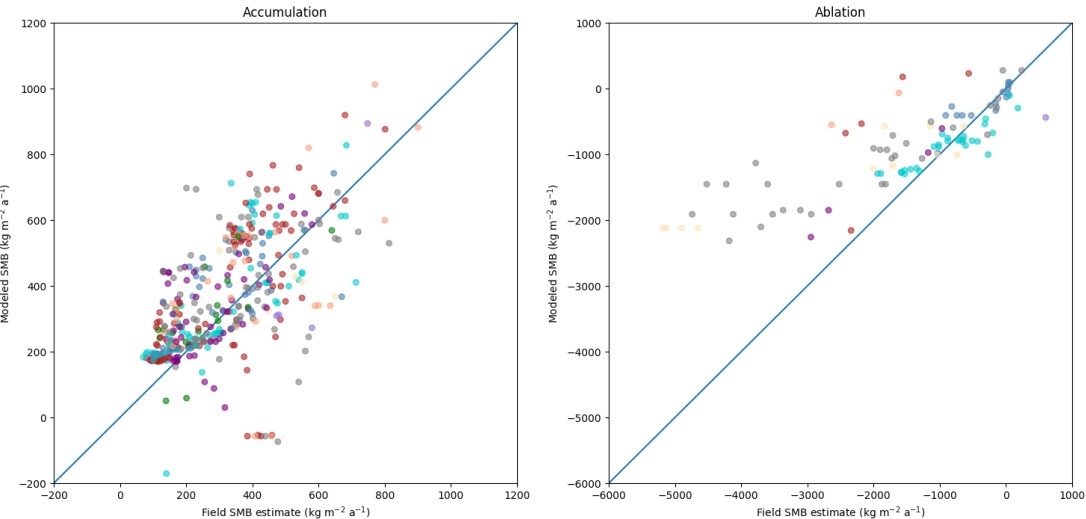

**Figure 14.** CESM annual SMB versus observed estimates at locations taken from accumulation (left) and ablation zones (right). Observational data were compiled as detailed in section 2. In both figures, colors correspond to the eight major drainage basins, as shown in Fig. 1, and the 1:1 line is drawn in blue.

that CESM is currently not able to capture the SMB well in the ablation areas of basin 6 (gray, the southwest GrIS). Figure 13 shows this bias as well. Separating ablation from accumulation provides the source of bias: the overly positive SMB in region 6, and overall, is due to too little ablation relative to observations, not too much accumulation.

While SMB comparisons by basin are helpful, model biases in elevation can provide clues as to the source. Figure 15
presents CESM and observed SMB over accumulation and ablation regions versus their elevation, and colored by basin, so that model developers are able to identify areas where there is an elevation mismatch. Because the model and observed data are not co-located, their horizontal locations are different. The observed data show a strong positive linear relationship between SMB and elevation up to about 1500 m, above which almost all points (except several in basin 1) have a positive SMB. The model is also able to capture this characteristic, however the linear relationship is more diffuse, and the linear increase in ablation with
decreasing elevation is too weak in CESM in basin 6 of the GrIS. Referring back to Fig. 6, the temperature gradient from the higher central part of GrIS to the edges is slightly weaker in CESM than RACMO2.3 in summer (and even more so in winter, not shown), with colder temperatures at the edges, which is consistent with the slope of elevation versus SMB shown in Fig. 15.

LIVVkit breaks down the model versus elevation data comparison along several key transects in the ablation zone, and
Fig. 16 shows that for CESM, all 4 transects contain some mismatch, but the Qammanarssup Sermia and Kangerlussuaq (K)-transects (basin 6) echo the overly weak modeled slope of elevation versus SMB ablation as seen in Fig. 15. As for the accumulation zone, LIVVkit's processed IceBridge transect data is presented as contours by location in the GrIS along with

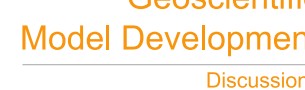
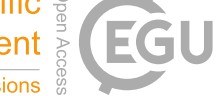



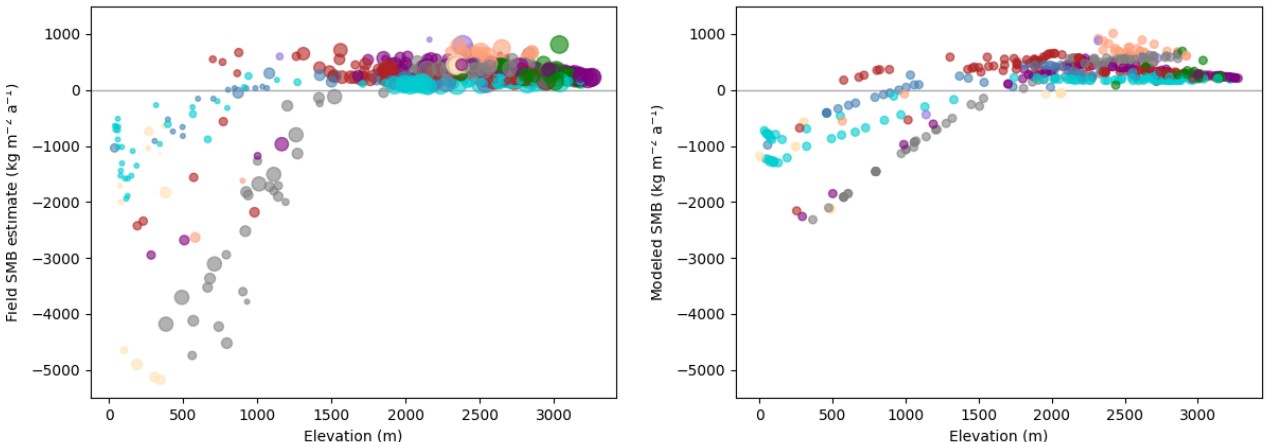

**Figure 15.** Field estimates of annual SMB as a function of observed elevation (left) and CESM annual SMB as a function of model elevation (right) from both accumulation and ablation zones. For the observed data, the size of the point represents the number of years in the field record, with larger points containing comparatively more temporal information than smaller points, and each point represents an annual surface mass balance estimate averaged across at least one year of data. For the modeled data, each point represents a model cell containing an available field estimate location. In both figures, colors correspond to the drainage basins as shown in Fig. 1.

model versus observed data in Fig. 17. The CESM bias has a latitudinal gradient; SMB is overestimated (e.g., basin 2) compared to annual IceBridge estimates in the Northern half of the GrIS and the bias decreases going South, becoming too low relative to the observed data in the Southern half. The source of this bias could be analyzed within the context of temperature, cloud cover, precipitation, and latent and sensible heat fluxes using figures such as 8 and 5, but covering the relevant seasons, but that
5   is beyond the scope of this survey.

## 5   Conclusions

An initial capability to perform automated validation of land ice and coupled models is presented, and we encourage the land ice modeling and larger computational Earth sciences community to bring additional tools and analysis to improve the software from this baseline. For example, generalization of the software to use with other models and the inclusion of more data are top
10   priorities. Although LIVVkit targets some aspects of relevant variables that affect land ice models within the coupled Earth system model as detailed above, there are additional analyses that could deliver useful information for both model developers and analysts. One example is to provide seasonal and long term SMB trends for information about model stability and forcings. Specifically, seasonal (summer) SMB estimates relative to the PROMICE dataset, which we currently only show as annualized





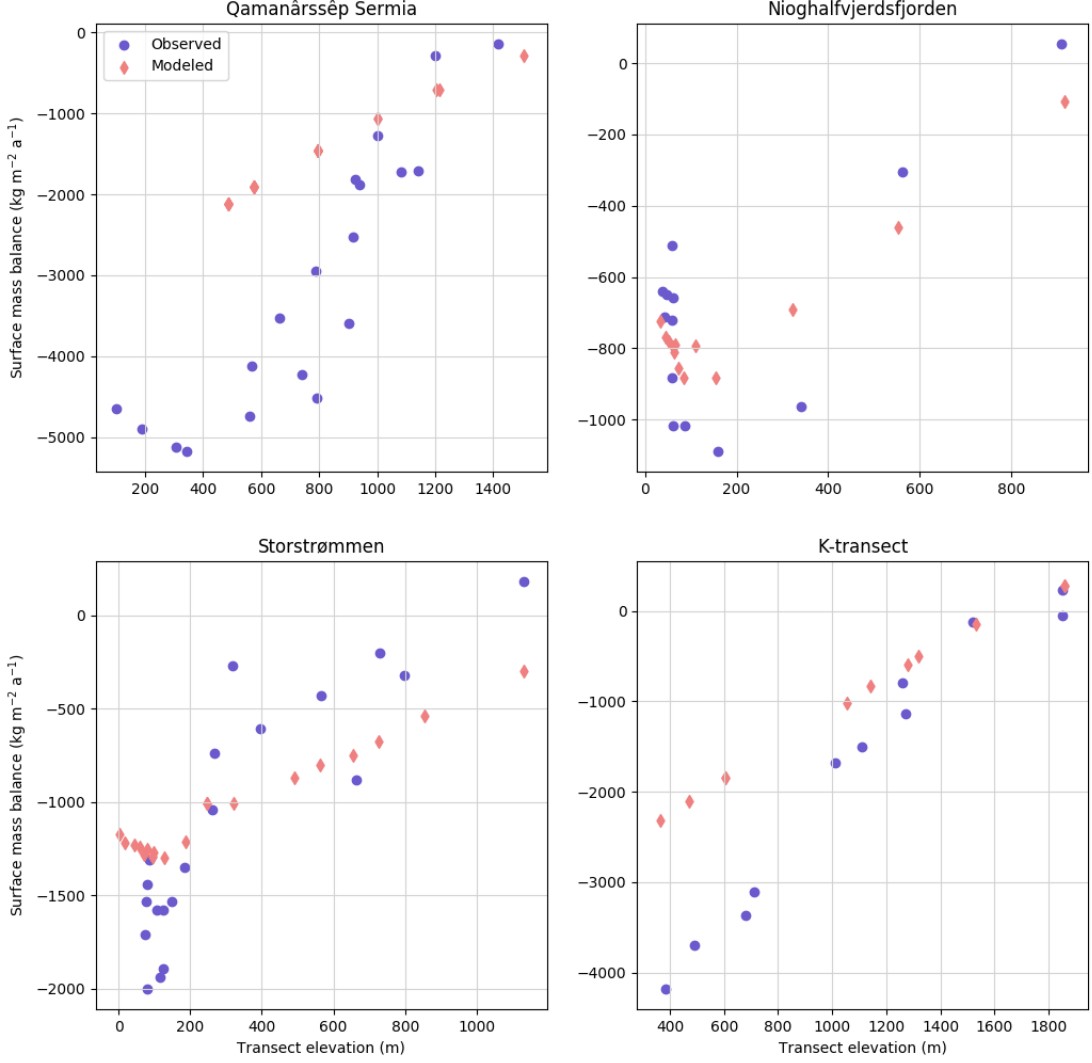

**Figure 16.** Annual surface mass balance as a function of observed elevation at four different areas in the ablation zone of GrIS. Red dots are modeled SMB and elevation, while blue dots are observed SMB and elevation from pit/core field estimates. Observation data were compiled from the PROMICE database (Machguth et al., 2016).





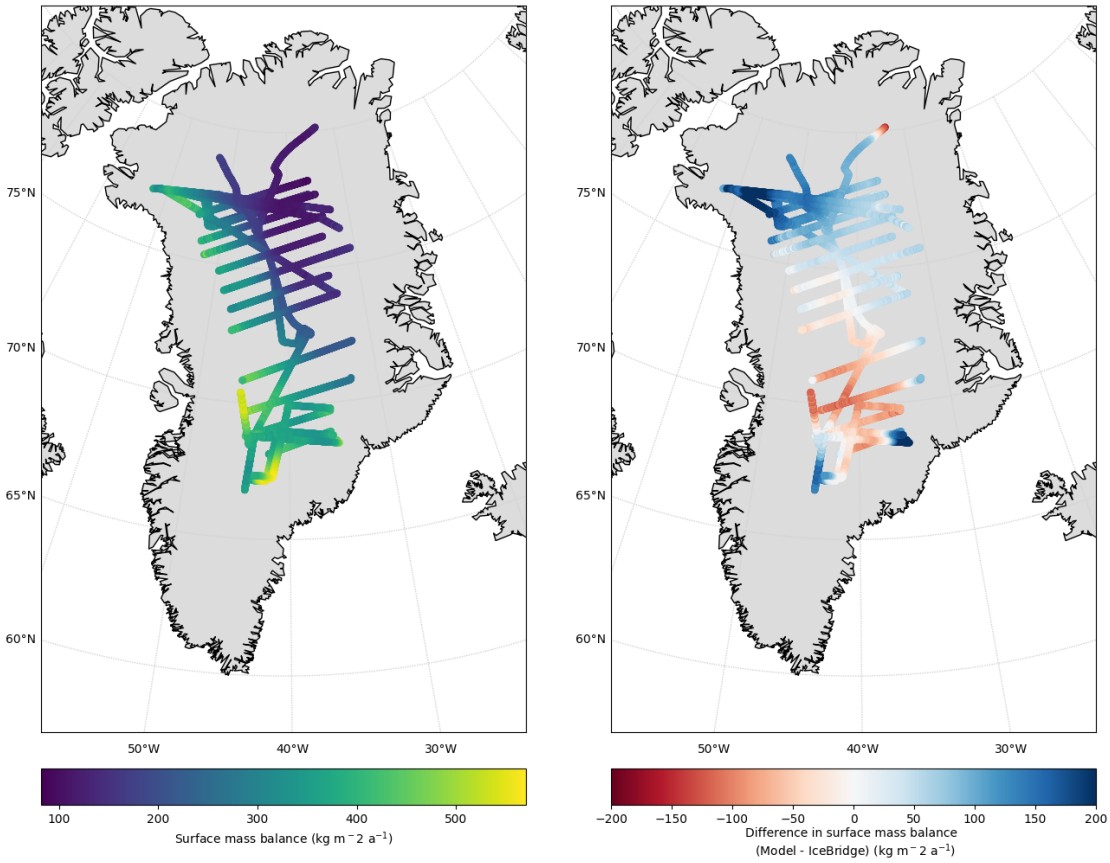

**Figure 17.** Observed annual surface mass balance of GrIS along IceBridge altimetry transects (left) and the differences between the observations and model along the transects (right).

values, could be extended. Extensions to assess the Antarctic ice sheet (AIS) is also greatly needed and are a near term priority. The recent release of Quantarctica, version 3[3], provides a collection of ice cores to consider for inclusion in LIVVkit.

Beyond new capabilities, additional observational data from both current and new sources would provide more information and build model confidence. For example, the LandSat8 GoLIVE global ice velocity data derived from Landsat 8[4] with 300 m
5   spacing, error and quality parameters, and subannual, and in some cases monthly or less, time periods. A recent synthesis of modeled and remotely-sensed inferences of the basal thermal state of GrIS (MacGregor et al., 2016) would provide a starting point for comparison temperature data. Additional SMB data to consider include data values from ASIRAS airborne radar and neutron-probe density measurements (Overly et al., 2016) and longer records from specific region to better analyze time series behavior, e.g. van de Wal (2012). Machguth et al. (2016) provide a history of SMB observations as a "state-of-the-art" that can
10   be used as a baseline for extension. Another source for additional SMB data, the Surface Mass Balance and Snow on Sea Ice

---

[3]https://www.pgc.umn.edu/news/quantarctica-version-3-released/
[4]https://nsidc.org/data/golive



Working Group (SUMup, 2018), compiles flights and ice snow pit/ice core data that includes includes PARCA cores that were included in LIVVkit as mentioned above as well as aerial flights from both Greenland and Antarctica. The IceBridge dataset also included in LIVVkit uses the more recent 2013-2014 flights, however earlier flights as detailed in Koenig et al. (2016) would provide more comparison within the ablation zone.

5    Next steps for LIVVkit being pursued include a connection to the CmCt validation framework that compares ice sheet models to altimetry and gravimetry satellite observations from the Ice, Cloud, and land Elevation Satellite (ICESat) and Gravity Recovery and Climate Experiment (GRACE) (Price et al., 2017). The substantial postprocessing required to compare the model with observations would complement the other metrics presented here, as an opt-in feature. Performance validation, which would expand from an initial computational verification capability as presented in Kennedy et al. (2017), is underway; the goals are to track model computational behavior on high performance computers. Efforts to enable LIVVkit's kernels, the Extended V&V for Earth Systems (EVE), to handle ensembles of output and provide sensitivies of variables to perturbed initial conditions are also being pursued, targeting verification using short ensembles of climate model configurations (e.g. (Mahajan et al., 2017)). Future work to develop and deploy uncertainty quantification techniques using LIVVkit within a larger ensemble based workflow would enable more robust uncertainty information regarding climate projections.

15    Several challenges exist in deploying LIVVkit for large-scale multimodel validation. There are several efforts we recommend for improved verification and validation in general, and to make LIVVkit more robust and extensible. Using common model projections and data conventions would allow many different models to participate in LIVVkit post processing with minimal code changes and other related information such as external mask and areal information etc. Some accepted protocol should be developed for LIVVkit to postprocess the data automatically. The land ice community is a relative newcomer within the coupled Earth system model community, so adoption of already accepted formats in use by other components would facilitate this transition. In any case, the breadth of ice sheet model development has created an opportunity to provide critically important simulations to the climate community, and this validation package is a step toward a predictive capability for land ice models, both on their own and coupled to a global Earth system model.

*Code and data availability.* The LIVVkit version 2.1 source code and documentation, which includes the process to procure and process the surface mass balance data, are available via a modified BSD license at https://github.com/LIVVkit/LIVVkit and https://livvkit.github.io/Docs/, respectively. The reader can access the preprocessed data from the simulations, with instructions, at https://code.ornl.gov/LIVVkit/lex, to use LIVVkit produce the validation output shown here. Refer to (Price et al., 2017), (Vizcaíno et al., 2013), and (Nöel et al., 2015) for instructions to procure the CISM-A, CESM, and RACMO simulation data we used as illustration. The comparison data for ice sheet validation can be accessed at (`jhkennedy/cism-data`; Kennedy, 2017) and refer to the Climate Data Guide (Pincus, R. and NCAR Research Staff (Eds)., 2016) for the atmosphere reanalysis fields. More details about data and code access are provided in section 2.

*Competing interests.* There are no competing interests of which the authors are aware.

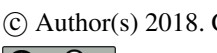



*Acknowledgements.* Support for this work was provided through Scientific Discovery through Advanced Computing (SciDAC) program funded by the U.S. Department of Energy Office of Advanced Scientific Computing Research and Office of Biological and Environmental Research. This manuscript has been authored by UT-Battelle, LLC and used resources of the National Center for Computational Sciences at Oak Ridge National Laboratory, both of which are supported by the Office of Science of the U.S. Department of Energy under Con-

5    tract No.DE-AC05-00OR22725. Contributions of CSZ made possible by support from DOE ACME DE-SC0012998 and NASA ACCESS NNX14AH55A. The United States Government retains and the publisher, by accepting the article for publication, acknowledges that the United States Government retains a non-exclusive, paid-up, irrevocable, world-wide license to publish or reproduce the published form of this manuscript, or allow others to do so, for United States Government purposes. The software described in this document is available through a modified BSD license accessible at https://github.com/LIVVkit/LIVVkit.git.





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
