# Peer review of "LIVVkit 2.1: Automated and extensible ice sheet model validation"

_Geoscientific Model Development, 2018_

## Short Comment (SC1) · 27 Apr 2018

Could you please make sure that the provided DOI links to the correct version 2.1 in the title and also add a reference in the manuscript? Many thanks.

Lutz Gross GMD Executive Editor

---

## Referee Comment (RC1) · Anonymous Referee #1 · 25 May 2018

The paper entitled "LIVVkit 2.1: Automated and extensible ice sheet model validation" by Katherine Evans et al. presents recent developments of LIVVkit, a verification/validation package for stand-alone ice sheet models and coupled Earth System models. They present here how it can be used to validate a Greenland ice sheet simulation using CISM-Albany and CESM. An analysis of the comparison between the modeled surface mass balance (SMB) and RACMO's SMB, for example, shows that Greenland's southwest coast has insufficient ablation.

[Figure]

**1  General Comments**

Overall the manuscript is well written, easy to read, with appropriate figures. What I am concerned about is the novelty: LIVVkit has already been described in ample detail in JAMES about a year ago (Kennedy et al 2017). It does not seem to me, and I could be wrong, that the extension to new datasets required significant code development as the package was already developed and ready to ingest new datasets (see Kenedy et al. 2017). The analyses shown here are interesting, and provide some good illustrations as to how LIVVkit can be used to determine biases and evaluate the overall performance about a standalone ice sheet model or a coupled ice/climate model, but I don't think this is within the scope of GMD.

What's also not very clear here, is how this package could be used by other modeling frameworks, like other standalone ice sheet models (e.g., PISM, Elmer, ISSM, etc) or other climate models. The package seems to be exclusively developed for CISM-Albany and CESM as it can only deal with a certain type of input file (that is not described by the way) on specific supercomputers.

**2  Specific comments**

- p2 l6: increased outflow → increased ice discharge ?
- p2 l8: Earth system models (no s in system)
- p2 l22 and elsewhere: consider changing "predictions" to "projections"
- p4 l3: I would disagree with this statement. It was probably true 5 or 6 years ago but the community now has access to a lot of time series (surface velocities, surface elevations, ice retreat) that are not used to initialize numerical ice sheet models.

- p12 l2: $L_2 \rightarrow L^2$

- p25 l4: LandSat8 → Landsat 8

- Figure 2c, 4c, 5c and 6c: these figures are difference plots and it would be much better to visualize the difference if the color bar was centered around 0, using a different color scheme such as blue-white-red.

- p30 l 12: CMCt → CmCt

**3  References**

Kennedy, J. H., et al. (2017), LIVVkit: An extensible, python-based, land ice verification and validation toolkit for ice sheet models, J. Adv. Model. Earth Syst., 9, 854-869, doi: 10.1002/2017MS000916.

---

## Referee Comment (RC2) · Anonymous Referee #2 · 6 Jun 2018

1) The acronym LIVVkit is used many times, in the abstract, title, and body, but the name is never written out in long form, so the reader cant know what its supposed to mean.

2) p2 l13: the acronym ESM is put after the second instance of using the term Earth system model, but should be after the first instance on line 8.

3) p4 l9: The citation for jhkennedy/cism-data uses a different font then the rest of the paper and citations, this should be changed to the same font used elsewhere. This same inconsistency shows up in the citation on p26 l29.

4) p4 l10: The method described for obtaining the LIVVkit data, "a host of publicly available sources" is described in the LIVVkit documentation "Verification" section as the following wget command: "wget jhkennedy.org/sites/default/files/LIVVkit/livvkit2.0.0_example_dataset.tar.gz" this command returns a 404 not found error. The data link should be fixed in the repo, or the citation here should be changed to point to the correct current location.

5) p6 l18: Instead of "It does this," change this to "LIVVkit does this."

6) p8 l17: The paper claims that "LIVVkit provides a set of single execution, task-parallel postprocessing (bash) scripts" but after looking though the github repo and documentation I couldn't find anything like this. Please provide a citation of where to find these scripts.

7) p8 l26: NCO works with non-gridded data as well.

8) p9 l21: LCF should be pluralized.

9) Figures: I would suggest a different color pallet, for example selecting one that is able to be easily read by the color blind. See http://www.somersault1824.com/tips-for-designing-scientific-figures-for-color-blind-readers/ The color pallet used in fig5 is much better then the one used for fig2 and fig4

10) p15: The spacing table2 and the text below it should be increased to match fig5 and its corresponding text.

11) There's inconsistency on how RACMO version 2.3 is denoted throughout the paper. In places its called "RACMO version 2.3" in others "RACMO2.3" or "RACMO 2.3"

12) p21: the pagination marker for page 21 is on top of the text for fig13. The figure should be raised up so they dont overlap each other.

---

## Author Comment (AC1) · 2 Jul 2018

Overall reply:

We thank the reviewers for taking valuable time to review our paper. Their comments have been incorporated into an updated draft that we have provided with the replies. They have improved the manuscript.

Reviewer1

1 General Comments

"Overall the manuscript is well written, easy to read, with appropriate figures. What I am concerned about is the novelty: LIVVkit has already been described in ample

detail in JAMES about a year ago (Kennedy et al 2017). It does not seem to me, and I could be wrong, that the extension to new datasets required significant code development as the package was already developed and ready to ingest new datasets (see Kennedy et al. 2017). The analyses shown here are interesting, and provide some good illustrations as to how LIVVkit can be used to determine biases and evaluate the overall performance about a standalone ice sheet model or a coupled ice/climate model, but I don't think this is within the scope of GMD."

Thank you for your comments. Although the main software infrastructure to perform verification and a proof-of-concept of validation as laid out in detail in Kennedy et al. (2017) existed, the additional software and testing required to perform each validation piece presented in the paper is non-trivial. Refer to section 3 for the details. The confusion may lie with the fact that the software to perform all the preprocessing for LIVVkit to present the analysis, occurs within a sub-repository named "LEX" (Kennedy and Evans, 2018) and is currently model specific because the ice sheet community does not always follow Climate and forecast (CF) metadata conventions that other model components require for model intercomparisons protocols (MIPs). We have updated the paper on pg 8 L7-9 and L13-19 to make this point more clearly.

There is also novelty in the breadth and scale of validation and comparison to observational data. We are not aware of any comparisons to renalaysis cloud data over the Greenland ice sheet over the same period of record with a coupled model with active ice sheet, nor are we aware of an SMB comparison of this breadth to observational data as shown here. This is also within scope for GMD, since one aim and scope bullet they provide is "new methods for assessment of models, including work on developing new metrics for assessing model performance and novel ways of comparing model results with observational data." The ability to process data of the scale of 1km resolution over Greenland is also unique in the scale of computing the software is able to handle. So there is novelty in the software, the use of large scale computing facilities, and the scientific analysis.

"What's also not very clear here, is how this package could be used by other modeling frameworks, like other standalone ice sheet models (e.g., PISM, Elmer, ISSM, etc) or other climate models. The package seems to be exclusively developed for CISM-Albany and CESM as it can only deal with a certain type of input file (that is not described by the way) on specific supercomputers."

Although we illustrate the software's utility to perform evaluations using example output from specific models, any model that produces netcdf files with the necessary variables, metadata, and masking/interpolation files can be processed by LIVVkit, with updates to the postprocessing. As ice sheet modeling groups move toward the goal of CF convention (refer to Goelzer et al., 2018), the ability to extend LIVVkit will become easier. It is true that the largest files from a 1km resolution ice sheet model (here from CISM-Albany) can only be processed with the largest computing capabilities, but models are being developed for these scales so we feel it's important to provide the capability to process the data for those who have access to it. Those who do not can still execute the software to target lower resolution output on smaller computing systems, a point now mentioned on pg8 L27.

Developing a LIVVkit Extension was under-described in LIVVkit's documentation. We've updated the documentation and you can see a more detailed description of how to add an extension here: https://livvkit.github.io/Docs/lex.html#developing-a-custom-extension. A new LIVVkit extension only requires providing a python file with a 'run(name, config)' function, and a JSON file which provides the path to the aforementioned python file to LIVVkit. Any script developed by a modeling group for analysis can simply be wrapped in that run function and executed within LIVVkit. With the software additions in LIVVkit and LIVVkit/LEX, it should be straightforward for users to add new methods for ice sheet or coupled model analysis within LIVVkit 2.1, as we mention pg11 L4-5.

2 Specific comments

1) p2 l6: increased outflow -> increased ice discharge

Change made to match Church et al. (2013) phrasing.

2) p2 l8: Earth system models (no s in system)

Change made, page 2 line 6

3) l22 and elsewhere: consider changing "predictions" to "projections"

Agreed. We have incomplete information to make predictions, so changing to "projections" in paragraph starting on page 2, line 9.

4) p4 l3: I would disagree with this statement. It was probably true 5 or 6 years ago but the community now has access to a lot of time series (surface velocities, surface elevations, ice retreat) that are not used to initialize numerical ice sheet models.

We should mention that part of this issue is due to, in part, because current initialization techniques use either a data assimilation method where much of the most informative observational data for model initialization (as with the CISMA case we use here) or a free spin-up technique which often produces a significant mismatch between the models and observations (Goelzer et al., 2018). Therefore, LIVVkit targets models that use the first approach, where validation is most relevant. We have updated the statement to reflect this nuance (pg 4 L1-3). There is also a mismatch between what models can currently simulate and available data. For example, full continent land ice models that include ice retreat will benefit from comparisons to observed retreat. That said, it is true that there are new campaigns to gather data that will be very useful for modelers and therefore good candidates for inclusion within updates of LIVVkit. We mention this in the conclusion section (paragraph beginning pg 26 L1).

5) p12 l2: L2 -> L2

Change made.

6) p25 l4: LandSat8 -> Landsat 8

Change made.

7) Figure 2c, 4c, 5c and 6c: these figures are difference plots and it would be much better to visualize the difference if the color bar was centered around 0, using a different color scheme such as blue-white-red.

We have updated the figures in the paper, as well as the others in the software that are not shown in the paper, to match your suggestion.

8) p30 l 12: CMCt -> CmCt

Change made.

Reviewer 2

1) The acronym LIVVkit is used many times, in the abstract, title, and body, but the name is never written out in long form, so the reader cant know what its supposed to mean.

Yes, good catch, we have added the definition in both the abstract and the introduction (pg 2 L33)

2) p2 l13: the acronym ESM is put after the second instance of using the term Earth system model, but should be after the first instance on line 8.

Change made.

3) p4 l9: The citation for jhkennedy/cism-data uses a different font then the rest of the paper and citations, this should be changed to the same font used elsewhere. This same inconsistency shows up in the citation on p26 l29.

The additional name of the software is not really necessary, so we just provided the cite.

4) p4 l10: The method described for obtaining the LIVVkit data, "a host of publicly available sources" is described in the LIVVkit documentation "Verification" section as the following wget command: "wget jhkennedy.org/sites/default/files/LIVVkit/livvkit2.0.0_example_dataset.tar.gz" this command returns a 404 not found error. The data link should be fixed in the repo, or the citation here should be changed to point to the correct current location.

Our apologies, this now been corrected in the documentation and LIVVkit's README on github. Unfortunately, Google has recently changed their file hosting policies for security purposes, so users will no longer be able to use wget to download the file on the command line. Instructions have been added that describe how to download the testing and reference datasets used in the verification example, which will have to be done through a web-browser for the time being.

5) p6 l18: Instead of "It does this," change this to "LIVVkit does this."

Change made.

6) p8 l17: The paper claims that "LIVVkit provides a set of single execution, taskparallel postprocessing (bash) scripts" but after looking though the github repo and documentation I couldn't find anything like this. Please provide a citation of where to find these scripts.

These were in a branch of LIVVkit, which is not ideal. So we have moved them to LEX (Kennedy and Evans, 2018) in the \postproc\MODELNAME directory, and we now explain how postprocessing the model data to prepare it for LIVVkit works within LEX in the README https://code.ornl.gov/LIVVkit/lex/tree/master/postproc and corresponding model type subdirectory's READMEs. In the paper on pg 8 L13-19, we point to the user to the LEX repository to extend the scripts for other models for LIVVkit.

7) p8 l26: NCO works with non-gridded data as well.

Correct, we changed it to "...programs that operate on a diversity of scientific data."

8) p9 l21: LCF should be pluralized.

Change made.

9) Figures: I would suggest a different color pallet, for example selecting one that is able to be easily read by the color blind. See http://www.somersault1824.com/tips-fordesigning-scientific-figures-for-color-blind-readers/ The color pallet used in fig5 is much better then the one used for fig2 and fig4

These have been updated for the figures in the paper and for plots made with the software to be more more color blind aware and also consistent (e.g. 0 line is white)

10) p15: The spacing table2 and the text below it should be increased to match fig5 and its corresponding text.

We have added the caption above the table, which gives it good spacing, and is apparently the typical location.

11) There's inconsistency on how RACMO version 2.3 is denoted throughout the paper. In places its called "RACMO version 2.3" in others "RACMO2.3" or "RACMO 2.3"

Now, the "RACMO version 2.3 (RACMO2.3)" is introduced as such the first time (pg 5 L8) then its referred to as RACMO2.3 thereafter.

12) p21: the pagination marker for page 21 is on top of the text for fig13. The figure should be raised up so they dont overlap each other.

Change made.

Citations within replies to reviewers above: Kennedy, J. H. and Evans, K. J.: LIVVkit/LEX: LIVVkit Extensions, https://code.ornl.gov/LIVVkit/lex, upcoming 0.1 release, 2018.

Goelzer et al. (2018). Goelzer, H., Nowicki, S., Edwards, T., Beckley, M., Abe-Ouchi, A., Aschwanden, A., Calov, R., Gagliardini, O., Gillet-Chaulet, F., Golledge, N. R., Gregory, J., Greve, R., Humbert, A., Huybrechts, P, Kennedy, J. H., Larour, E., Lipscomb, W. H., Le clec'h, S., Lee, V., Morlighem, M., Pattyn, F., Payne, A. J.,

Rodehacke, C., Rückamp, M., Saito, F., Schlegel, N., Seroussi, H., Shepherd, A., Sun, S., van de Wal, R., and Ziemen, F. A.: Design and results of the ice sheet model initialisation experiments initMIP-Greenland: an ISMIP6 intercomparison, The Cryosphere, 12, 1433-1460, https://doi.org/10.5194/tc-12-1433-2018, 2018.

Please also note the supplement to this comment:
https://www.geosci-model-dev-discuss.net/gmd-2018-70/gmd-2018-70-AC1-supplement.pdf

**Supplement:**

**LIVVkit 2.1: Automated and extensible ice sheet model validation**

Katherine J. Evans[1], Joseph H. Kennedy[1], Dan Lu[1], Mary M. Forrester[2], Stephen Price[3], Jeremy Fyke[3,a], Andrew R. Bennett[4], Matthew J. Hoffman[3], Irina Tezaur[5], Charles S. Zender[6], and Miren Vizcaíno[7]

[1]Computational Earth Sciences Group, Oak Ridge National Laboratory, Oak Ridge, TN, USA
[2]Colorado School of Mines, Golden, CO, USA
[3]Fluid Dynamics and Solid Mechanics Group, Los Alamos National Laboratory, Los Alamos, NM, USA
[4]U. Washington Dept. of Civil and Environmental Engineering
[5]Sandia National Laboratories, Albuquerque, NM, USA
[6]Departments of Earth System Science and Computer Science, University of California, Irvine, USA
[7]Institute for Marine and Atmospheric Research, Utrecht University, Utrecht, Netherlands
[a]now with Associated Engineering, Vernon, BC, Canada

**Correspondence:** K. J. Evans (evanskj@ornl.gov)

**Abstract.** A collection of scientific analyses, metrics, and visualizations for robust validation of ice sheet models is presented using the *Land Ice Verification and Validation toolkit* (LIVVkit), version 2.1. This software collection targets stand-alone ice sheet or coupled Earth system models, and handles datasets and operations that require high-performance computing and storage. LIVVkit aims to enable efficient and fully reproducible workflows for post-processing, analysis, and visualization of

5    observational and model-derived datasets in a shareable format, whereby all data, methodologies, and output are distributed to users for evaluation. *Extending from the initial LIVVKit software framework, w*e demonstrate  *a diverse collection of* Greenland ice sheet simulation *validation metrics* using the coupled Community Earth System Model, CESM, as well as an idealized stand-alone high-resolution ice sheet model, CISM-Albany. As one example of the capability, LIVVkit analyzes the degree to which models capture the surface mass balance (SMB) and identifies potential sources of bias, using

10    recently available in-situ and remotely sensed data as comparison. Related fields within atmosphere and land surface models, e.g. surface temperature, radiation, and cloud cover, are also diagnosed. Applied to the CESM1.0, LIVVkit identifies a positive SMB bias that is focused largely around Greenland's southwest region that is due to insufficient ablation.

*Copyright statement.* This manuscript has been authored by UT-Battelle, LLC under Contract No. DE-AC05-00OR22725 with the U.S. Department of Energy. The United States Government retains and the pub- lisher, by accepting the article for publication, acknowledges that

15    the United States Government retains a non-exclusive, paid-up, irrevocable, worldwide license to publish or reproduce the published form of this manuscript, or allow others to do so, for United States Government purposes. The Department of Energy will provide public access to these results of federally sponsored research in accordance with the DOE Public Access Plan (http://energy.gov/downloads/doe-public-access-plan).

[revised manuscript text omitted]
. *Some aspects of the postprocessing scripts to execute step (1) are model specific because ice sheet and coupled model configurations with an active ice models do not always follow Climate and Forecast (CF) conventions for metadata nor produce the same type of variables and corresponding units. To enable the scripts to target output from other models, the user would*
15   *alter the scripts that target the models presented here, provided within a LIVVkit Extensions (LEX) git repository (Kennedy and Evans, 2018). Updating the scripts to update the years of simulation and/or variables available to postprocess are expected to be frequent and is a straightfoward process at the top of the scripts. To extend the scripts to process output from models that have different resolutions, units of measure, and/or ice sheet masking, the user needs to alter the remapping scripts provided to interpolate the observed data to match the model. Details about this process are provided with the scripts in the repository*.

[revised manuscript text omitted]

---

## Author Response (AR2)

**Detailed response to Editor comments**

Manuscript #: gmd-2018-70

**Editor comments:**

I have now looked at your manuscript, the previous JAMES manuscript and your response to reviewers in details. It took me quite a bit of time since it may indeed be questioned whether there is enough new aspects to warrant publication in GMD, as highlighted by one of the reviewers. In the end, I think this is the case.

However, I still would like to have some small corrections & improvements to the manuscript regarding the data used. We try at GMD, as I am sure you are aware, to improve traceability of datasets and software. In your current version this is a quote from your data access:

> LIVVkit produce the validation output shown here. Refer to (Price et al., 2017), (Vizcano et al., 2013), and (Nel et al., 2015) for instructions to procure the CISM-A, CESM, and RACMO simulation data we used as illustration.

From a quick reading, this does not provide any leads on what is the actual access policy of the datasets used.

Price et al. 2017:

> The Community Ice Sheet Model code is available at http://oceans11.lanl.gov/cism/index.html (Price et al., 2014; Lipscomb et al., 2017). For the Albany momentum balance solver, please see the code availability statement in Tezaur et al. (2015a). The raw ICESat and GRACE data discussed above are available for download at http://nsidc.org/data/icesat/data.html and http://podaac.jpl.nasa.gov/datasetlist?search=GRACE, respectively. The CmCt on-line service is available at https://ggsghpcc.sgt-inc.com/cmct/ and the CmCt source code will be made available upon request. Model forcing and initial condition datasets are available through direct contact with the respective authors (see Sect. 2.3).

This does not discuss output datasets except for GRACE, but this is not the subject of the paper nor your reference.

Vizcaino 2013 does not have a data usage section, nor any license on use I could find. No access methodology either.

Nöel 2015 has no indication on data sharing or access.

Therefore and in short, you are indicating readers to look for papers where little is explained on how to access the data and nothing is written on the licensing.

This is not appropriate for GMD. I would like therefore an update of this with precise references to the datasets, licensing and access possibilities there.

**Author response:**

Thank you for your comments. Although we cannot control the data access policies of previously published works, nor feasibly distribute their raw datasets which are on the order of hundreds of GBs, we now discuss these issues and our strategy to overcome them in the manuscript. We also state that we provide all the data and code necessary to reproduce all these analyses and that they are available via LIVVkit v2.1.6 (DOI:10.5281/zenodo.1322735) and LEX v0.1.0 (DOI:10.5281/zenodo.1937579). We have updated our data access statement to read:

> The LIVVkit version 2.1 source code and documentation (Kennedy et al., 2018a) are available via a modified BSD license at https://github.com/LIVVkit/LIVVkit and https://livvkit.github.io/Docs, respectively. The reader can reproduce all the analyses presented here with LEX (Kennedy et al., 2018b), which is available via a modified BSD license at https://code.ornl.gov/LIVVkit/lex, and is documented in the LIVVkit documentation. More details about data and code access are provided in sections 2 and 3, respectively.

where Kennedy et al. (2018a) and Kennedy et al. (2018b) are citations of the above DOIs, respectively. For each dataset provided in LEX, there is a README.md file that discusses the source of the raw data, how to acquire the raw data, the processing steps that were used to produce the processed data, and the code used to process the data. This strategy should be more than adequate to satisfy GMD's data and software traceability and access policies.

In the same vein, we improved our description of LEX, which is entirely new and unpublished work, and we have differentiated it from LIVVkit (which has also been expanded since the JAMES article was published). Therefore, we have updated their descriptions in the manuscript and, most significantly, reorganized and expanded parts of Section 3 "Software infrastructure for validation."

Finally, we also made a several small changes, namely fixing capitalization inconsistenties of "LIVVkit" and following the recommended style for "in situ" throughout the manuscript.

You can see all of our changes in the included difference PDF, where additions are shown in blue, and subtractions are shown in red.

**LIVVkit 2.1: Automated and extensible ice sheet model validation**

Katherine J. Evans[1], Joseph H. Kennedy[1], Dan Lu[1], Mary M. Forrester[2], Stephen Price[3], Jeremy Fyke[3,a], Andrew R. Bennett[4], Matthew J. Hoffman[3], Irina Tezaur[5], Charles S. Zender[6], and Miren Vizcaíno[7]

[1]Computational Earth Sciences Group, Oak Ridge National Laboratory, Oak Ridge, TN, USA
[2]Colorado School of Mines, Golden, CO, USA
[3]Fluid Dynamics and Solid Mechanics Group, Los Alamos National Laboratory, Los Alamos, NM, USA
[4]U. Washington Dept. of Civil and Environmental Engineering
[5]Sandia National Laboratories, Albuquerque, NM, USA
[6]Departments of Earth System Science and Computer Science, University of California, Irvine, USA
[7]Institute for Marine and Atmospheric Research, Utrecht University, Utrecht, Netherlands
[a]now with Associated Engineering, Vernon, BC, Canada

**Correspondence:** K. J. Evans (evanskj@ornl.gov)

**Abstract.** A collection of scientific analyses, metrics, and visualizations for robust validation of ice sheet models is presented using the Land Ice Verification and Validation toolkit (LIVVkit), version 2.1, and the LIVVkit Extensions repository (LEX), version 0.1. This software collection targets stand-alone ice sheet or coupled Earth system models, and handles datasets and  analyses that require high-performance computing and storage. LIVVkit aims to enable efficient and fully repro-
5   ducible workflows for post-processing, analysis, and visualization of observational and model-derived datasets in a shareable format, whereby all data, methodologies, and output are distributed to users for evaluation. Extending from the initial  LIVVkit software framework, we demonstrate Greenland ice sheet simulation validation metrics using the coupled Community Earth System Model, CESM, as well as an idealized stand-alone high-resolution ice sheet model, CISM-Albany. As one example of the capability, LIVVkit analyzes the degree to which models capture the surface mass balance (SMB) and identifies
10   potential sources of bias, using recently available  in situ and remotely sensed data as comparison. Related fields within atmosphere and land surface models, e.g. surface temperature, radiation, and cloud cover, are also diagnosed. Applied to the CESM1.0, LIVVkit identifies a positive SMB bias that is focused largely around Greenland's southwest region that is due to insufficient ablation.

*Copyright statement.* This manuscript has been authored by UT-Battelle, LLC under Contract No. DE-AC05-00OR22725 with the U.S.
15   Department of Energy. The United States Government retains and the publisher, by accepting the article for publication, acknowledges that the United States Government retains a non-exclusive, paid-up, irrevocable, worldwide license to publish or reproduce the published form of this manuscript, or allow others to do so, for United States Government purposes. The Department of Energy will provide public access to these results of federally sponsored research in accordance with the DOE Public Access Plan (http://energy.gov/downloads/doe-public-access-plan).

[revised manuscript text omitted]